

# Technical note: Evolution of convective boundary layer height estimated by Ka-band continuous millimeter wave radar at Wuhan in central China

Zirui Zhang[1,2,3]   Kaiming Huang[1,2,3]   Fan Yi[1,2,3]   Fuchao Liu[1,2,3]   Jian Zhang[4]   and

Yue Jia[5]

[1]School of Electronic Information, Wuhan University, Wuhan, China

[2]Key Laboratory of Geospace Environment and Geodesy, Ministry of Education, Wuhan, China

[3]State Observatory for Atmospheric Remote Sensing, Wuhan, China

[4]School of Geophysics and Geomatics, China University of Geosciences, Wuhan, China

[5]NOAA Chemical Sciences Laboratory, Boulder, CO, USA

Email Address: hkm@whu.edu.cn

Zip code: 430072





**Abstract.** Using the vertical velocity (VV) observed by a Ka-band millimeter wave cloud radar (MMCR)
at Wuhan, we investigate the evolution of convective boundary layer height (CBLH) based on a specified
threshold of VV variance. Compared with the CBLH retrieved from the lidar range corrected signal (RCS),
the MMCR-derived CBLH exhibits lower values for a few hours post-sunrise and pre-sunset, but outside
these two periods, they are generally in good agreement. Relative to the lidar RCS that is susceptible to the
historical aerosol mixing processes, the CBLH estimated from the MMCR VV variance shows a rapid
response to thick clouds and a less contamination by aerosol residual layer and long-distance transport of
sand and dust, thus the MMCR VV observation can capture the CBLH evolution very well. The MMCR
observation in 2020 depicts the seasonal and monthly variations in the CBLH. The seasonal mean CBLH
reaches the peak heights of 1.29 km in summer, 1.14 km in spring, and 0.6 km in autumn and winter, with
occurrence time between 13:30 and 15:00 LT. The maximum (mean) value of mean (daily maximum) CBLH
rises steadily from 0.66 (0.87) km in January to 1.47 (1.76) km July, followed by a gradual decline to 0.42
(0.5) km in December. Statistical standard deviations are monthly-dependent, indicating the significant
influence of weather conditions on the CBLH. This study improves our understanding of the Ka-band
MMCR's capability to monitor the CBLH, emphasizing its utility in tracking the dynamical processes in the
boundary layer.
1. **Introduction**
Atmosphere boundary layer is located in the lowermost layer of the troposphere, and directly impacts the
air-land/sea interaction because of its link between the surface and the free atmosphere (Stull, 1988). Owing
to the combined effects of gravity, viscosity, and friction of the ground and uneven temperature distribution
caused by radiation, the boundary layer is characterized by complex dynamical processes, with the
prominent turbulence features of vorticity and compressibility, especially during daytime (Bernardini et al.,



2012; Schneider, 2008). Typically, the boundary layer top varies diurnally following the local surface
temperature with a magnitude from a few tens of meters to several kilometers. The convective boundary
layer (CBL) is a type of atmosphere boundary layer driven primarily by convection. After sunrise, the
Earth's surface absorbs solar shortwave radiation, which increases upward sensible heat, enhancing the
near-surface convection and elevating the CBL heights (CBLH) gradually (LeMone et al., 2010; Grossman
and Robert, 2005; Yates et al., 2001). In the afternoon, as the sensible heat flux decreases, turbulent activity
is weakened, causing the CBL to contract downward. Generally, the CBL collapses after sunset, and aerosol
particles within the CBL are transformed into a residual layer. The residual layer descends gradually due to
the sinking effect until it is mixed with the CBL driven by the next day's post-sunrise convection
(Blay-Carreras et al., 2014; Heus et al., 2010; Tennekes and Driedonks, 1981). Since the boundary layer
controls the exchanges of heat, momentum, moisture and mass between the ground and the free atmosphere
(Mahrt, 1999; Holtslag and Nieuwstadt, 1986), the structure of boundary layer is an important input variable
in numerical weather-prediction and climate models (Edwards et al., 2020).

The evolution of CBLH has a distinct daily cycle, which is dominated mainly by surface sensible heat

from solar radiation, and can be influenced significantly by weather and local topography (Kwon et al., 2022;
Ribeiro et al., 2018). Typically, the CBL is capped by a stable temperature inversion layer, constraining the
upward development of convection, thus under the circumstances, the inversion layer bottom is often
identified as the top of boundary layer (Stull, 1988). At the height of CBL top, turbulent mixing weakens
markedly, leading to substantial changes and strong vertical gradient of atmospheric parameters, such as
potential temperature, relative humidity and aerosol concentration. Consequently, the CBL top determines to
a great extent the vertical dispersion of aerosol particles (Kong and Yi, 2015; Pal et al., 2015; Stull, 1988).
Thus, the CBLH plays a crucial role in air quality and atmospheric environment evaluations, as the
concentration of surface emissions and pollutants is closely related to the CBLH (Li et al., 2017; Tang et al.,

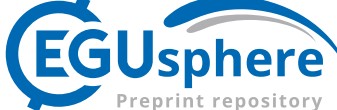

2016; Liao et al., 2015; Liu and Liang, 2010; Seibert, 2000). Besides, at the boundary top, moisture,
aerosols and other chemical substances are entrained to the free atmosphere, creasing an entrainment
transition zone between the boundary layer and the free atmosphere (Franck et al., 2021; Liu et al., 2021;
Brooks and Fowler, 2007). Hence, the CBLH also plays an important role in influencing the cloud formation
and precipitation above the CBL through regulating water vapor and aerosols (as condensation nuclei)
entrained into the free atmosphere (Guo et al., 2017; Brooks and Fowler, 2007; Neggers et al., 2004; Brown
et al., 2002).

The observations of in situ radiosonde and remote sensing are extensively used to estimate the CBLH

and its seasonal feature in different geographical environments. The radiosonde data have a widely
geographical distribution and long-term accumulation, which is convenient to study the climatology of
boundary layer in different regions. Meanwhile, radiosonde can obtain many meteorological parameters
with high precision, such as pressure, temperature, relative humidity, and horizontal wind velocity and
direction, providing a possibility to retrieve the boundary layer height through different algorithms (Seidel et
al., 2010; Seibert, 2000). Typically, the vertical gradients of potential temperature and water vapor
(including relative humidity and specific humidity) are used to determine the CBLH (Zhang et al., 2022;
Guo et al., 2021; Dang et al., 2019; Liu and Liang, 2010; Seidel et al., 2010; Stull, 1988). Additionally, the
boundary top can be evaluated using the profiles of refractivity and bulk Richardson number derived from
the temperature, pressure, vapor pressure and horizontal wind data (Burgos-Cuevas et al., 2021; Guo et al.,
2016; Zhang et al., 2014; Seidel et al., 2012; Basha and Ratnam, 2009). These retrieval algorithms provide
insights into the features of boundary layer from the perspective of energy exchange, mass transport,
turbulent motion and effect on radio propagation. Even so, radiosonde faces a severe limitation to capture
the clear development of CBL due to its conventional release schedule, which typically occurs only twice a
day.



In contrast to radiosonde, ground-based remote sensing offers high temporal resolution in observational
profiles, which is essential to investigate the diurnal evolution of boundary layer. Wind profile radar can
obtain the atmospheric wind speed and direction by decomposing the Doppler shift of electromagnetic
waves backscattered by the vertical inhomogeneity of atmospheric refractive index structure constant due to
the gradients in temperature and relative humidity, and the fluctuation of refractive index caused by
turbulence (Liu et al., 2020; Singh et al., 2016; Seibert, 2000). In this way, some parameters from the wind
profile radar measurement, such as signal-to-noise ratio, Doppler spectral width and refractive index
structure constant, are utilized to retrieve the height of boundary layer for every 30-60 min based on their
vertical gradients or chosen thresholds (Burgos-Cuevas et al., 2023; Bianco et al., 2022; Solanki et al., 2021;
Liu et al., 2020; Allabakash et al., 2017; Sandeep et al., 2014). Nevertheless, previous studies showed that
the top of CBL derived from the radar observation may be influenced by strong residual layer and shallow or
large entrainment zone (Sandeep et al., 2014; Bianco and Wilczak, 2002).
Lidar is regarded as a powerful detection equipment for capturing boundary layer development due to its
high sensitivity to echo signals from various atmospheric components. Its relatively short operating
wavelength allows it to receive echoes backscattered not only from aerosol and cloud particles, but also
from atmospheric molecules. Nevertheless, since Rayleigh scattering of atmospheric molecules is much
weaker than Mie scattering of aerosol particles, the profile of lidar backscatter coefficient or range corrected
signal (RCS) from aerosols is extensively used to determine the CBLH by tracing the height of aerosol
concentration plunge. Accordingly, many techniques have been developed to identify the extreme value of
RCS gradient (Liu et al., 2021; Su et al., 2020; Dang et al., 2019; Yang et al., 2017; Kong and Yi, 2015;
Granados-Muñoz et al., 2012). As a simplified low-power lidar, ceilometer is initially designed to measure
the height of cloud base, thus similarly, the backscatter profile from the ceilometer observation can be
applied to the CBL investigation (Zhang et al., 2022; Schween et al., 2014; Van Der Kamp and McKendry,





2010). However, limited by the ability of lasers to penetrate clouds, the CBLH may be contaminated and
even misinterpreted by clouds within the boundary layer in the lidar and ceilometer measurements (Schween
et al., 2014).

With the advances in atmospheric sounding technology, the vertical velocity from Doppler lidar provides

a direct estimation of convectively driven boundary layer, which can reduce the impact of strong aerosol
concentration within the residual layer on the retrieved CBLH (Burgos-Cuevas et al., 2023; Dewani et al.,
2023; Huang et al., 2017; Schween et al., 2014; Barlow et al., 2011). At the initial stage of CBL formation in
the morning and the rapid decline stage of CBL in the afternoon, aerosol particles in the residual layer may
cause the CBLH to be overestimated by about several hundred meters. This discrepancy often reflects the
historical effect of aerosol mixing rather than the current situation of convectively driven turbulence
(Burgos-Cuevas et al., 2023; Schween et al., 2014; Pearson et al., 2010). In the Doppler lidar observation, a
specified threshold of vertical velocity variance is used to define the height of CBL top. This method has
been validated as reliable by comparison with the measurements from other equipment, and the sensitivity
of threshold has been discussed across different sites (de Arruda et al., 2018; Manninen et al., 2018;
Schween et al., 2014; Barlow et al., 2011; Pearson et al., 2010). In contrast to lidar with large blind range
and limited penetrating cloud capability, microwave cloud radar offers good low altitude coverage and
superior performance in cloud penetration. In the cloud observation, there always exist a weak echo layer
near the surface, from which the vertical velocity can be retrieved. However, there are few reports on the use
of vertical velocity obtained from Doppler cloud radar for the CBL investigations.

In present study, we estimate the CBLH based on the vertical velocity from a Ka-band millimeter wave

cloud radar (MMCR) at Wuhan, and analyze the evolving features of CBL in different seasons by using the
observational data with high temporal resolution. In section 2, the instruments and their data are briefly
described, followed by the methodology in section 3. In section 4, we present four examples of CBLH



diurnal evolution in different seasons by comparing the CBL tops identified from the MMCR and lidar
measurements, and then investigate the monthly and seasonal characteristics of CBLH over Wuhan in
Section 5. Section 6 provides a summary.

**2. Instruments and Data**
In this study, the CBLH derived from the MMCR observation is compared with that from the lidar
measurement. The Ka-band MMCR and lidar are situated at the Atmospheric Remote Sensing Observatory
(ARSO) in Wuhan University (30.5°N, 114.4°E). MMCR antenna is positioned 40 m above sea level, which
is about 30 m lower than lidar telescope. Wuhan, an inland megacity in central China, is located in the east
of Jianghan Plain, with a resident population of over 12 million. The city is dominated by the subtropical
monsoon humid climate, which is characterized with by abundant precipitation and four distinct seasons
(Guo et al., 2023). Due to heavy traffic and industrial activities, large amounts of aerosols are emitted from
the industrialized metropolis. Meanwhile, sandstorms from the northwest often pass through Wuhan,
especially in spring. These sandstorms cause the remarkable variation in the spatial distribution and
concentration of aerosols. Frequent sand and dust activity along with cloudy weather poses significant
challenges for the Ka-band MMCR and lidar in accurately capturing the CBL evolution.
**2.1 Ka-Band Radar**
The MMCR established by the ARSO is a Ka-band frequency-modulated continuous wave (FMCW)
Doppler radar, which is shown in Figure 1. The radar adopts the mode of transmitting and receiving
separation through two same Cassegrain antennas with 1.5 m diameter. Radiation antenna transmits a mean
power of 50 W at operating frequency of 35.035 GHz through 0.38° width beam. Backscatter echoes from
aerosol and cloud particles are received by reception antenna, and then are sent to the signal processing
subsystem to obtain the radial distribution of parameters that represent the characteristics and motion of



particles, such as reflectivity factor, Doppler velocity, Doppler spectrum width, signal-to-noise ratio, and so
on. Because of almost continuous transmission and reception, FMCW radar has an adjustable range
resolution by modulating and demodulating the continuous wave, and a much higher duty cycle relative to
pulse radar, leading to a higher temporal resolution in the FMCW radar measurement. In non-precipitation,
the MMCR measurement has a time resolution of 0.26 s and a maximum unambiguous velocity of 4.30 m s$^{-1}$,
which are adjusted to be 0.104 s and 10.75 m s$^{-1}$ in precipitation as the size and falling speed of
hydrometeors increase (Mao et al., 2023), respectively. The MMCR observation has been applied to the
investigations of cloud and precipitation over Wuhan in previous works (Fang et al., 2023; Mao et al., 2023).

The MMCR has a maximum detectable distance of about 30 km and a sensitivity of -30 dBZ at the

distance of 10 km. In the MMCR measurement, there are weak echoes generally less than -40 dBz within a
few kilometers above the surface. The weak echoes near the surface are attributed to the backscattering of
plankton and insects in some studies (Franck et al., 2021; Chandra et al., 2010; Achtemeier, 1991), and are
also suggested to come from the scattering of dust particles in other studies (Görsdorf et al., 2015; Clothiaux
et al, 2000; Moran et al., 1998). Considering that the size of large dust particles, plant aerosol particles,
aerosol particles from combustion, and so on, can be much larger than 10 μm, it is possible for the large
aerosol particles to cause these weak echoes in the MMCR observation. The servo-mechanical subsystem
conducts the radar to work at specified directional mode or scanning mode. In 2020, the radar was operated
at the vertically pointing mode, and the observation is recorded with a vertical resolution of 30 m. In this
study, we attempt to explore the CBL evolution at Wuhan from the Ka-band MMCR observation in 2020.
**2.2 Polarization Lidar**

The polarization lidar developed by the ARSO is about 0.5 km away from the Ka-band radar. The lidar

transmits vertically the pulses of 120 mJ at operating wavelength of 532 nm with a repetition rate of 20 Hz
by a frequency-doubled Nd: YAG laser. The output polarized laser beam has a fine polarization purity with



183 depolarization ratio less than 1:10000 by using a Brewster polarizer. Light backscattered by aerosol and

184 cloud particles and atmospheric molecules is collected by a telescope with 0.3 m diameter. After separated

185 through an interference filter with 0.3 nm bandwidth centered at 532 nm, the elastically backscattered light

186 is incident on a polarization beam splitter prism, and then the two-channel polarized light are focused onto

187 two photomultiplier tubes (PMTs), respectively. The signals from the two PMTs are transferred to a personal

188 computer (PC)-controlled two-channel transient digitizer to obtain the echo signal intensity and volume

189 depolarization ratio through the PC processing. Backscatter coefficient are retrieved based on the backward

190 iteration algorithm under the condition of a given lidar ratio proposed by Fernald and Klett (Fernald, 1984;

191 Klett, 1981), and then the RCS and particle depolarization ratio are derived from the backscatter coefficient

192 and volume depolarization ratio (Freudenthaler et al., 2009; Immler and Schrems, 2003). Expanded laser

193 beam overlaps with the full field of view of receiving telescope at a height of 0.3 km, thus this height is the

194 low limit of lidar detection. The lidar data has a temporal resolution of 1 min, and a same vertical resolution

195 of 30 m as the MMCR data. The lidar configuration and depolarization comparison with the measurement

196 from the cloud-aerosol lidar and infrared pathfinder satellite observation (CALIPSO) were in detail

197 described in early study (Kong and Yi, 2015).

198  We regard the height of MMCR antenna as a baseline, thus considering that lidar telescope is about 30 m

199 higher than MMCR antenna, the initial height of lidar data is set at 0.33 km. Meanwhile, in the following

200 analysis, we use local time to represent time.

202 **3. Methodology**

203  In view of the CBLH derived from the vertical velocity (VV) in the MMCR observation but from the

204 RCS in the lidar measurement, we use different algorithms to determine the CBLH, respectively.

205 **3.1 Gradient, Variance and Wavelet Transformation Methods**



In the lidar observation, the gradient (Grd) method is often utilized to investigate the CBLH by
identifying the strongest or minimum gradient of RCS since the intensity of backscattered signal is
approximately proportional to the aerosol concentration (Kong and Yi, 2015; Lewis et al., 2013; Pal et al.,
2010; Emeis et al., 2008). The wavelet covariance transformation (WCT) method, with a chosen Harr
wavelet function, estimates the CBL top by investigating the correlation of the RCS variation with a step
function (Zhang et al., 2021; Angelini and Gobbi, 2014; Pal et al., 2010; Baars et al., 2008; Brooks, 2003).
Essentially, the WCT method can be considered as a smooth enhancement of Grd method, which may be
less affected by noise than the Grd method (Davis et al., 2000; Baars et al. al., 2008).
The Grd and WCT methods derive the CBLH from the change of echo signal intensity in the spatial
profile, while the variance (Var) method identifies the CBL top based on the variations of echo signal in the
temporal domain. The frequent exchange of matter and energy between the boundary layer and the free
atmosphere causes the dramatical variation of aerosol concentration on small time scales around the CBL
top. In this case, the height where the variance of backscattered signal reaches the maximum value is
regarded as the CBLH (Lammert and Bösenberg, 2006; Martucci et al., 2004; Piironen and Eloranta, 1995).
We estimate the CBLH from the lidar RCS every 30 min by using the three methods, and then the obtained
height is marked at the central time of 30 min.
**3.2 Threshold Method**
The VV variance is representative of the level of turbulent activity, thus a threshold of VV variance is
applied to determining the CBLH in the Doppler lidar measurement. The threshold is chosen to be 0.04 $m^2$
$s^{-2}$ in the regions with weak turbulence (Tucker et al., 2009), 0.3 $m^2 s^{-2}$ in a tropical rainforest (Pearson et al.,
2010), and 0.4 $m^2 s^{-2}$ in the region with central European climate (Schween et al., 2014; Träumner et al.,
2011), while the thresholds of 0.1 and 0.2 $m^2 s^{-2}$ are selected in the urban landscapes since the retrieved
CBLH is not heavily dependent on the given thresholds (Burgos-Cuevas et al., 2023; Huang et al. 2017;



Barlow et al., 2011). Similarly, the threshold method is also used to determine the CBLH from the VV
variance in the MMCR measurement, with a same duration of 30 min as the lidar observation.

Figure 2 presents the VV from the Ka-band MMCR observation and RCS (in arbitrary unit) from the

lidar measurement on 15 August 2020. By taking observations for 30 min from 11:45 to 12:15, we calculate
the mean VV and RCS, and estimate the position of CBL top by means of different algorithms, which are
shown in Figure 3. From the lidar RCS, the CBLH is 1.35 km in the Grd and WCT methods, and 1.32 km in
the Var method, indicating the consistent results for the three algorithms. In the MMCR observation, the VV
variance has a clear downward trend with height increasing, with the values of about 1.36 $m^2 s^{-2}$ from the
near ground to 0.15 $m^2 s^{-2}$ at 1.47 km, and then maintains slight fluctuations around the value of 0.15 $m^2 s^{-2}$
to higher altitudes. For a specified threshold of 0.3 $m^2 s^{-2}$, the CBL top is identified at the height of 1.35 km,
which is in good agreement with the lidar results. It can be noted from Figures 3d and 3f that the CBLHs in
the mean RCS profile are around the position with the most rapid change, while the CBLH retrieved from
the MMCR VV variance, representing the convectively driven turbulences, is not related to the vertical
variation of mean VV. Hence, the good consistency of CBLH derived from the MMCR and lidar
demonstrates that the MMCR VV variance is a fine proxy in the estimation of CBLH.

As shown in Figure 3e, the variance decreases quickly from 0.4 $m^2 s^{-2}$ at 1.29 km to 0.15 $m^2 s^{-2}$ at 1.47

km, indicating that the CBH top at noon is less sensitive to the selected threshold within 0.15-0.4 $m^2 s^{-2}$.
Figure 4 depicts the CBLHs on 15 August 2020 at the thresholds from 0.2 to 0.45 $m^2 s^{-2}$. Overall, the CBL
top declines with the increasing threshold, nevertheless, the CBLH from 09:30 to 17:30 remains relatively
stable with little change at the different thresholds. The discrepancy under these thresholds arises mainly in
the initial formation and final dissipation stages of CBL due to the large variabilities of turbulences with
time and space. Even so, the CBL has an approximately same initial (final) height of about 0.09 (0.12) km at
06:00 (21:00). In following analysis, we take 0.3 $m^2 s^{-2}$ as the threshold to determine the CBLH in the





MMCR observation.
**4. Case Investigation and Comparison**
Figure 5 presents the CBLH evolution on 15 August 2020 from the lidar RCS based on the Grd, Var and
WCT methods, and their comparison with that obtained from the MMCR VV variance, together with the
distribution of MMCR reflectivity factor in the range of 10-15 km. As shown in Figure 5c, due to the
influence of aerosol residual layer, the CBLH from the lidar RCS fluctuates from about 1.56 km at 06:00
down to 1.17 km at 09:30, however, with the sunrise at 05:50, the CBL top derived from the MMCR VV
variance gradually rises from about 0.09 km at 06:00 to 1.17 km at 09:30. It is interesting that the CBLH
from the lidar RCS variance drops at 07:30, and then shows a change similar to that from the MMCR VV
variance. When the CBL ascends gradually and mixes with the residual layer, the CBLHs in the lidar and
MMCR observations are consistent with each other between 09:30 and 17:00, including a slight drop at
12:30 and 14:30 (from the gradient and variance of RCS). The maximum height of CBL is about 1.71 km at
14:00 and 15:00 based on the VV variance and the RCS gradient and variance. One can note from the
reflectivity factor distribution in Figure 5b that cirrus clouds occur from 17:00, develop rapidly into the
thick clouds at about 11-14.4 km at 17:30, and then dissipate quickly after 17:30. In the MMCR observation,
the cirrus appearance makes a large contribution to a clear dip in the CBLH between 17:30 and 18:30,
nevertheless, the CBL top has a lift as the clouds dissipates rapidly, indicating that the convectively driven
turbulence and CBLH have an immediate response to radiation variation. The influence of clouds on the
CBLH is also reported in some earlier studies (Dewani et al., 2023; Bianco et al., 2022; Barlow et al., 2011).
The phenomenon of CBLH subsidence also arises in the lidar RCS, especially from the RCS variance, but
with a time lag due to the influence of historical mixing process on the aerosol distribution (Burgos-Cuevas
et al., 2023; Schween et al., 2014). After the sunset at 19:05, the CBLH retrieved by the VV variance drops
quickly to 0.27 km at 20:00 from 1.47 km at 19:00, while the top of aerosol residual layer (or horizontally



migrating aerosol layer) identified by the lidar stays at far higher level, in particular, from the RCS gradient
and WCT.

Next, we select the observations on 31 January, 12 November, and 19 March 2020 to compare the CBLH

evolutions. Figure 6 shows the CBLHs on 31 January derived from the four methods above, which are
overlaid on the MMCR VV and its variance and the lidar RCS, respectively. It is very cold in January at
Wuhan, and the weather is clear from the MMCR observation on 31 January, with a minimum (maximum)
temperature of -5 °C (4 °C) recorded in weather forecast. Owing to the convection inhibited largely by the
frigid surface and air, the VV variance shows that the CBLH develops very slowly upward to 0.3 km at
11:30 from 0.12 km at 07:30 as the sun rises at 07:15. Thereafter, the top of CBL climbs quickly to 0.9 km at
13:30, and reaches the maximum height of 0.99 km at 14:30, and during this period, the CBLH from the
lidar RCS experiences a similarly rapid uplift, and attains the peaks of 1.2 km at 14:00 from the RCS
gradient and variance, and 1.14 km at 14:30 from the RCS WCT. In addition, it can be seen from Figure 6d
that all the CBLH is slightly larger from the three RCS algorithms than from the VV variance. This implies
that a moderately smaller threshold may be appropriate for the estimation of CBLH in winter with weak
turbulence (Burgos-Cuevas et al., 2023; Huang et al., 2017; Tucker et al., 2009). After 14:30, the CBLH
from the VV variance descends gradually, and approaches the ground at 17:30 prior to the sunset at 17:57,
while at the sunset, the CBL top from the RCS is at 0.8-0.9 km due to the history of mixing processes.

Figure 7 presents the CBLHs determined from the MMCR and lidar observations on 12 November 2020.

With the sunset on this day in late autumn, the CBLH identified from the VV variance displays a little
fluctuation until 10:30. After then, the CBL is rapidly developed to 0.51 km at 11:30, and mixes fully with
the residual layer retrieved from the lidar RCS, thus the CBL tops have an approximately same evolution
between the MMCR and lidar observations from 11:30 to 17:30, with the maximum values of about
0.75-0.78 km at 15:00 and 16:00. As the sun goes down at 17:27, the CBL from the VV variance rapidly





shrinks close to the ground at 18:00, and aerosol particles left in the air form a residual layer, similar to the
two cases above.

Figure 8 depicts the CBLH variations in the MMCR and lidar observations on 19 March 2020, together

with the depolarization ratio from the lidar. In spring, sandstorms occur frequently in the northwest of China,
and sand and dust with different intensities are often blown to Wuhan. On this day, there is a fine sand and
dust layer mostly above 1.8 km, with the depolarization ratios of about 0.08-0.12 in Figure 8c, which can
also be noted from the MMCR VV distribution. Meanwhile, another sand and dust layer with the larger
depolarization ratios of about 0.14-0.16 passes through Wuhan from about 14:00, and mixes with the lower
part of the first sand and dust layer. In this situation, the MMCR observation indicates that the CBL starts to
develop gently upward from the sunrise, and the upward trend of CBLH is also presented in the lidar
measurement, but at higher altitudes. At 09:30, the CBLH is about 0.48 km in both the MMCR and lidar
observations, and then rises steadily to 1.32 km at 16:00 and 16:30, shows a good agreement between the
two observations. Subsequently, the CBLH from the VV variance undergoes two rapid declines. One occurs
from 1.2 km at 17:00 to 0.51 km at 18:00, which is probably related to the sand and dust deposition besides
the diminished radiation in the late afternoon, and the other arises after the sunset. However, because of the
effect of sand and dust, the CBLH from the lidar RCS increases slightly from 1.32 km at 16:30 to about 1.38
km at 18:00 and 18:30, and then decreases gradually with time.

The CBLH is identified through the spatial and temporal variation of aerosol concentration from the

lidar measurement but through the VV change in the time domain from the MMCR observation. The four
examples demonstrate that except for a few hours after the sunrise and before the sunset due to the influence
of aerosol residual layer, the CBL tops from the MMCR and lidar observations are in good agreement with
each other. The residual layer always causes a higher CBLH estimated by the lidar RCS than by the MMCR
VV because the convectively driven turbulence represented by the VV variance is less contaminated by the





residual layer. Hence, the MMCR VV observation can capture the CBLH evolution very well under a
threshold of VV variance, especially for the boundary layer in the blind range of lidar. In view of the
seasonal characteristics of convection, a slightly smaller threshold may be more suitable for the CBLH
estimation in winter with weaker turbulence. Owing to that the thermally driven convection is sensitive to
solar radiation, the CBL top identified from the VV variance has a swift response to clouds, which is
distinguished from the lidar observation due to the RCS affected by the history of aerosol mixing processes.

**5. Monthly and Seasonal Mean CBLHs**
To reveal the general characteristic of CBLH diurnal evolution in different months and seasons, we
calculate the monthly and seasonal mean CBLHs by using the MMCR VV on these days without
precipitation in 2020. Routinely, winter covers December, January, and February, and so on. Figure 9
illustrates the averaged CBLHs with the standard deviations superimposed on the mean VV variance in each
month and season. As we expect, the mean VV variance is the strongest in summer and the weakest in
winter. As the spot of direct sunlight slowly moves northward, the mean variance gradually increases from
January to July and August, and then decreases step by step from August to December. Interestingly, the
variance is significantly larger in spring than in autumn. These monthly and seasonal features of
convectively driven turbulence dominate the evolution of monthly and seasonal mean CBLHs. The
maximum height of CBL is 1.29 km at 14:30 and 15:00 in summer, 1.14 km at 13:30 in spring, and about
0.6 km at 13:30 and 14:00 in autumn and at 14:30 in winter. In summer, the CBLH displays a feature of
quick descent near twilight, and in autumn, the CBL shows a wider envelope with an earlier development
and a later dissipation relative to that in winter though their maximum CBLHs are almost the same.
Figure 10 presents the maximum value of monthly mean CBLH and corresponding occurrence time
from January to December. The maximum height rises steadily from 0.66 km in January to 1.47 km July and



1.44 km in August, and then drops gradually to the lowest altitude of 0.42 km in December. In weather
forecast record, there are 7, 3, 13, 3 and 0 days with moderate to heavy rain at Wuhan in September from
2018 to 2022, indicating that September 2020 is a rainy month. The MMCR observation also shows
frequently moderate to heavy rains for hours in September 2020, which may be responsible for an evident
reduction of CBL maximum height from August to September since a large latent heat flux due to the
evaporation on the surface can inhibit the development of thermally driven convection to a certain extent
(Dewani et al., 2023; Sandeep et al., 2014). The maximum height occurs is between 13:00 in November and
December and 17:30 in July. At Wuhan, the plum rain starts in June and prevails in July. As shown in Figure
9, the CBLH in July has the largest standard deviation (between 13:00 and 19:00) and the latest occurrence
time of maximum value over the whole year, which is possibly attributable to the cloudy and rainy weather
in addition to the strongest radiation. Similarly, the variability of weather conditions may be a major reason
why the maximum height arises 1-2 hours earlier in April-June than in March. Nevertheless, with the
gradual decline of solar radiation, the occurrence time of maximum height is steadily advanced from 17:30
in July to 13:00 in November and December.

Finally, we calculate the mean values and standard deviations of daily maximum CBLH and its

occurrence time in each month, which is presented in Figure 11. Figure 11 illustrates that the monthly mean
value of maximum CBLH has a variational trend similar to the maximum values of monthly mean CBLH in
Figure 10. The averaged maximum CBLH is raised from 0.87 km in January to 1.76 km July, and then
gradually decreases to the lowest altitude of 0.5 km in December, and its largest and smallest standard
deviations also arise in July and December, respectively. The occurrence time of averaged maximum CBLH
is the earliest at about 12:40 in December and the latest at 15:45 in August, which is slightly distinguished
from those in the maximum value of mean CBLH. The standard deviation of occurrence time is obviously
large in January, July and September. These results imply that the maximum height and its occurrence time



of daily CBL are significantly influenced by the weather conditions besides radiation since the VV variance
as a proxy of convectively driven turbulence is sensitive to the weather changes.

**6. Summary**

In this study, we investigate the diurnal evolution of monthly and seasonal mean CBLH at Wuhan by the

VV variance method based on the Ka-band MMCR observation, and compare the CBLH evolution with that
by the RCS gradient, variance and wavelet methods from the lidar measurement.

Using the MMCR VV observation on these days without precipitation in 2020, we statistically analyze

the monthly and seasonal variations of CBLH. The maximum value of monthly mean CBLH increases
steadily from 0.66 km in January to 1.47 km July and 1.44 km in August, and subsequently, decreases
gradually to the lowest height of 0.42 km in December. Analogously, the monthly mean value of daily
maximum CBLH rises from 0.87 km in January to 1.76 km July, and then gradually drops to the lowest
altitude of 0.5 km in December. The occurrence times is between 13:00 in November and December and
17:30 in July for the maximum value of monthly mean CBLH, but between about 12:40 in December and
15:45 in August for the monthly mean value of daily maximum CBLH, respectively. As for the seasonal
feature, the seasonal mean CBLH has the maximum heights of 1.29 km at 14:30 and 15:00 in summer, 1.14
km at 13:30 in spring, and 0.6 km at 13:30 and 14:00 in autumn and at 14:30 in winter. These results are
similar to those in early studies (Burgos-Cuevas et al., 2021; Guo et al., 2021; Solanki et al., 2021; Tang et
al., 2016; Kong and Yi, 2015). Meanwhile, the statistical standard deviations are monthly-dependent,
suggesting that the CBLH is not only regulated mainly by the solar radiation, but also affected significantly
by the weather conditions, such as clouds through decreasing radiation to the surface, and precipitation
through increasing the latent heat flux.

Besides the maximum values, the MMCR VV variance can capture the initial formation and final



dissipation stages of CBL very well relative to the lidar RCS. In the ascending and descending phases of
CBL, the CBLH from the lidar RCS is higher than from the MMCR VV variance, due to the high blind
range of lidar and the strong influence of aerosol residual layer on the lidar RCS. When the CBLH reaches
the height of CBL top identified by the lidar RCS, the CBL tops from the MMCR and lidar observations are
in good agreement until it is separated from the top of aerosol residual layer left behind in the late afternoon.
Additionally, in comparison to the lidar RCS affected by the history of aerosol mixing processes, the CBLH
in the MMCR observation shows a rapid response to clouds and a less contamination by the long-range
transport of sand and dust, indicating the efficiency of the VV variance method in the estimation of
convectively driven boundary layer, similar to in early studies (Dewani et al., 2023; Huang et al., 2017;
Barlow et al., 2011).
The case analysis indicates that the CBLH is not very sensitive to the VV variance thresholds of 0.2-0.45
$m^2 s^{-2}$, thus we chose a constant threshold of 0.3 $m^2 s^{-2}$ across all months and seasons. Whereas, the
investigation shows that a slightly smaller threshold may be more suitable for the weak convection in winter,
thus considering the seasonal and regional, and even weather characteristics of thermally driven convection,
the optimal threshold of VV variance in different scenarios require to be discussed carefully in the future. As
is known, the Ka-band MMCR is a powerful instrument for observing clouds and weak precipitation, thus
the MMCR measurement gives us an opportunity to study in detail the influence of clouds at different
heights on the CBLH and the CBLH evolution under different weather conditions.
In this case, the full-time and full-weather MMCR observation with low blind height can obtain the
whole evolution of CBLH in many weather conditions, which is helpful for us to gain an insight into the
CBL features and also provides the important input variables for weather-prediction and climate models.



**Code availability.** Software code to obtain the results is available upon request from the corresponding
author.
**Data availability.** All data used are available upon request from the corresponding author.
**Author contributions.** KH and FY conceptualized this study. ZZ and KH completed the analysis and the
manuscript. FL, JZ, YJ, and FY discussed the results and finalized the manuscript.
**Competing interests.** The authors declare that they have no conflict of interest.
**Financial support.** This work was supported by the National Key Research and Development Program of
China (2022YFB3901800 and 2022YFB3901805) and the National Natural Science Foundation of China

(42174189).

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



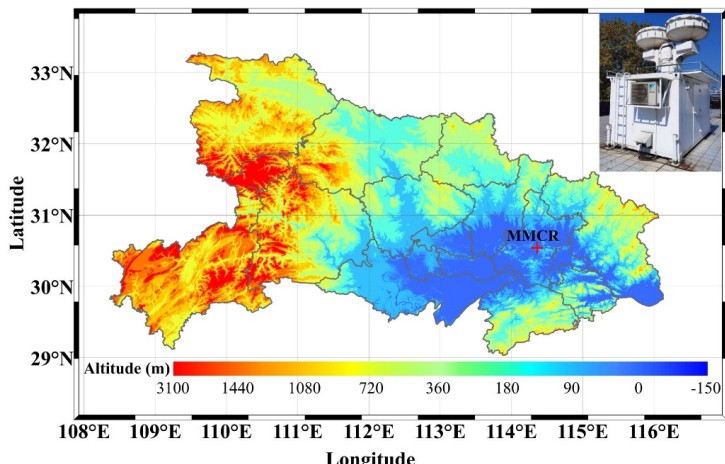


**Figure 1.** Topographic elevation map of Hubei Province and Ka-band MMCR located in Wuhan University

(30.54°N, 114.36°E). The red crisscross denotes the site of MMCR.



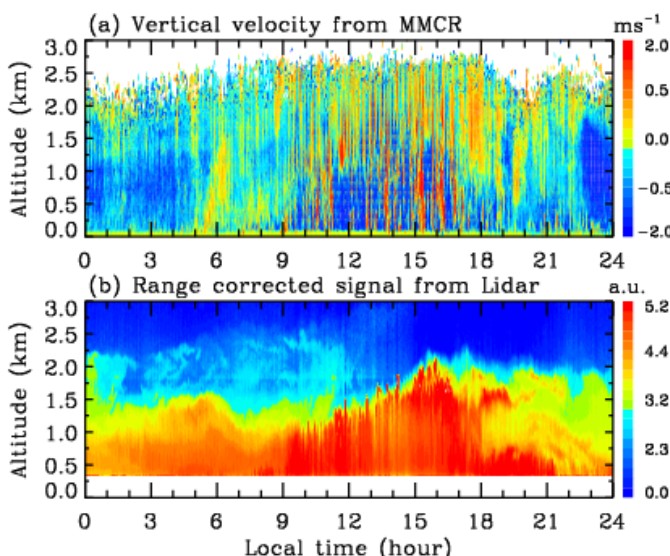


**Figure 2.** Time-height section of (a) vertical velocity from MMCR and (b) RCS from lidar on 15 August

2020.



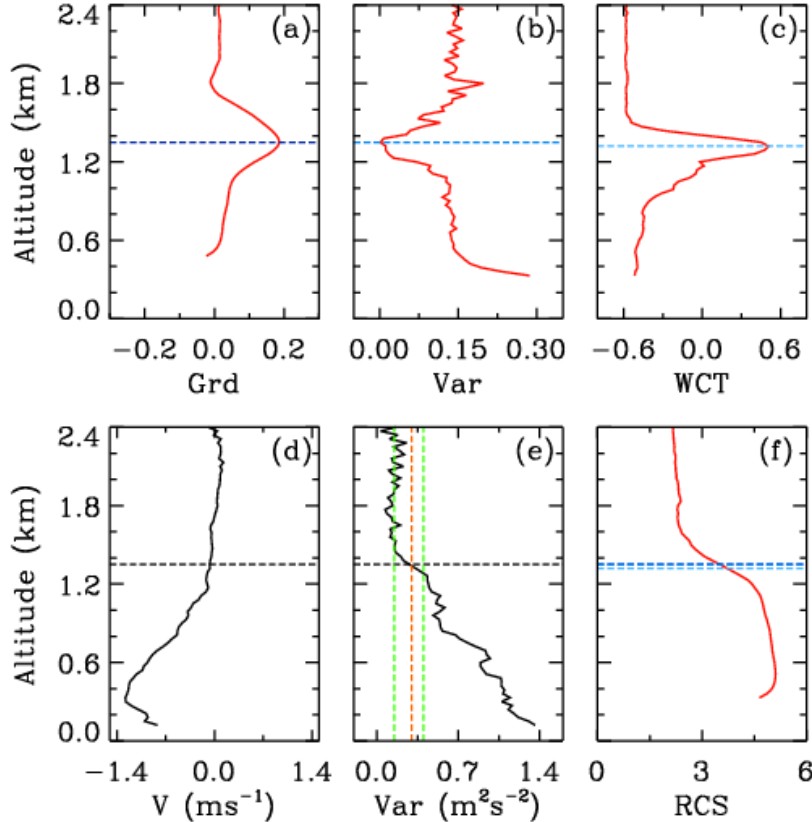


**Figure 3.** Profiles of (a) RCS gradient, (b) variance, (c) WCT and (f) RCS form lidar, and (e) vertical

velocity and (g) its variance from MMCR between 12:15 and 12:45 LT on 15 August 2020. In these panels,

the horizontal lines in different colors represent the CBLH determined by different methods. In Panel 3e, the

orange vertical line denotes the selected threshold of 0.3 $m^2 s^{-2}$, and the two green vertical lines correspond

to the variances of 0.15 and 0.4 $m^2 s^{-2}$, respectively.

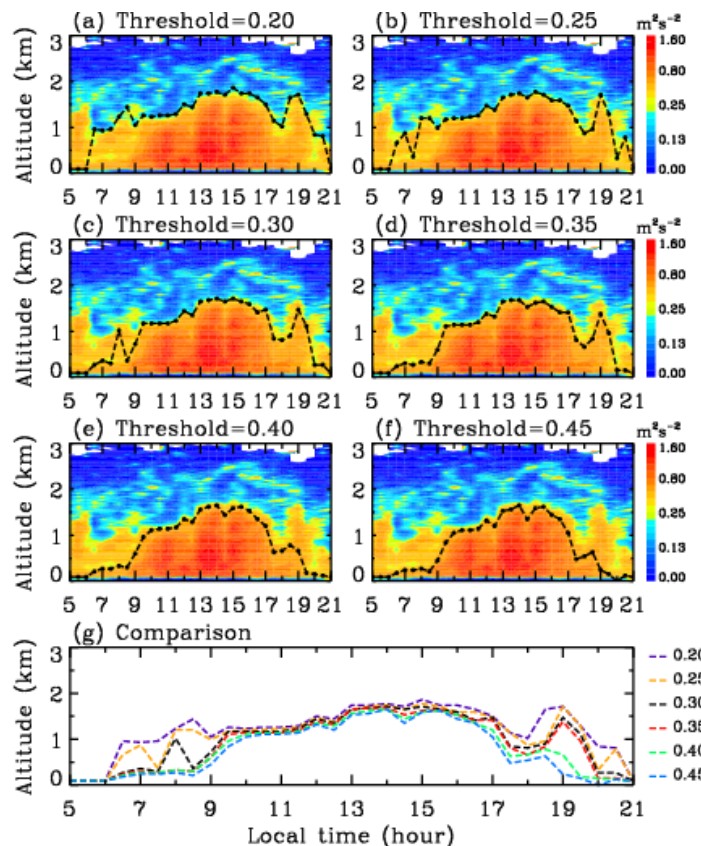

**Figure 4.** CBLHs derived from thresholds of (a) 0.2, (b) 0.25, (c) 0.3, (d) 0.35, (e) 0.4 and (f) 0.45 $m^2 s^{-2}$

superimposed over vertical velocity variance (color shading) from MMCR on 15 August 2020, and (g) their

comparison.

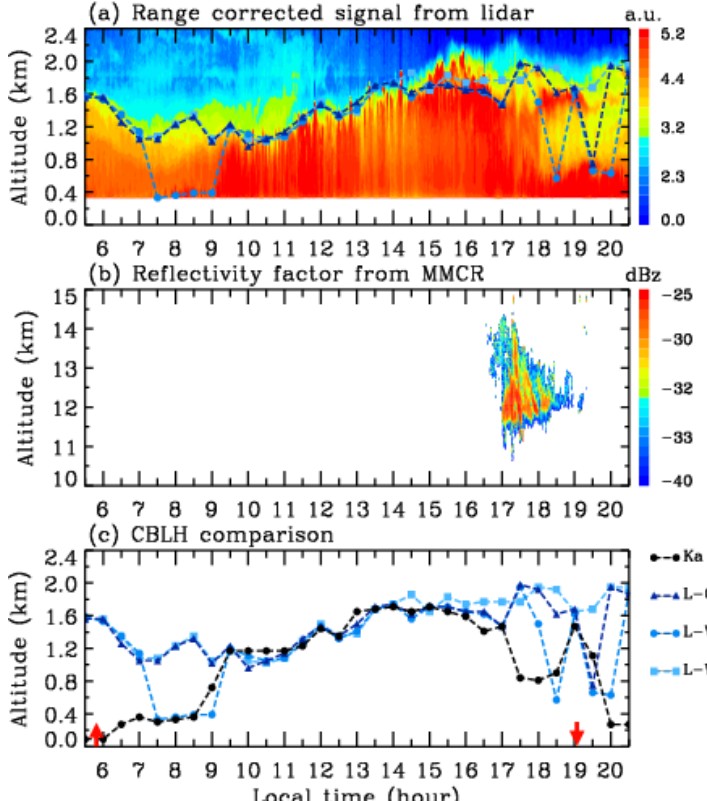

**Figure 5.** (a) Evolution of CBLH derived from RCS gradient, variance and WCT superimposed over lidar

RCS (color shading) on 15 August 2020, (b) reflectivity factor from MMCR, and (c) comparison of CBLHs

derived from MMCR and lidar observations. The black dash curve with circle (Ka) denotes the CBLH

determined by the variance threshold of 0.3 $m^2 s^{-2}$ in the Ka-band MMCR observation, while the dark blue,

blue and light blue dash curves with triangle (L-G), circle (L-V) and square (L-W) represent the CBLH

determined by the gradient, variance and WCT in the lidar measurement, respectively. In Panel 5c, the two

red arrows denote the time of sunrise and sunset at 05:50 and 19:05, respectively.



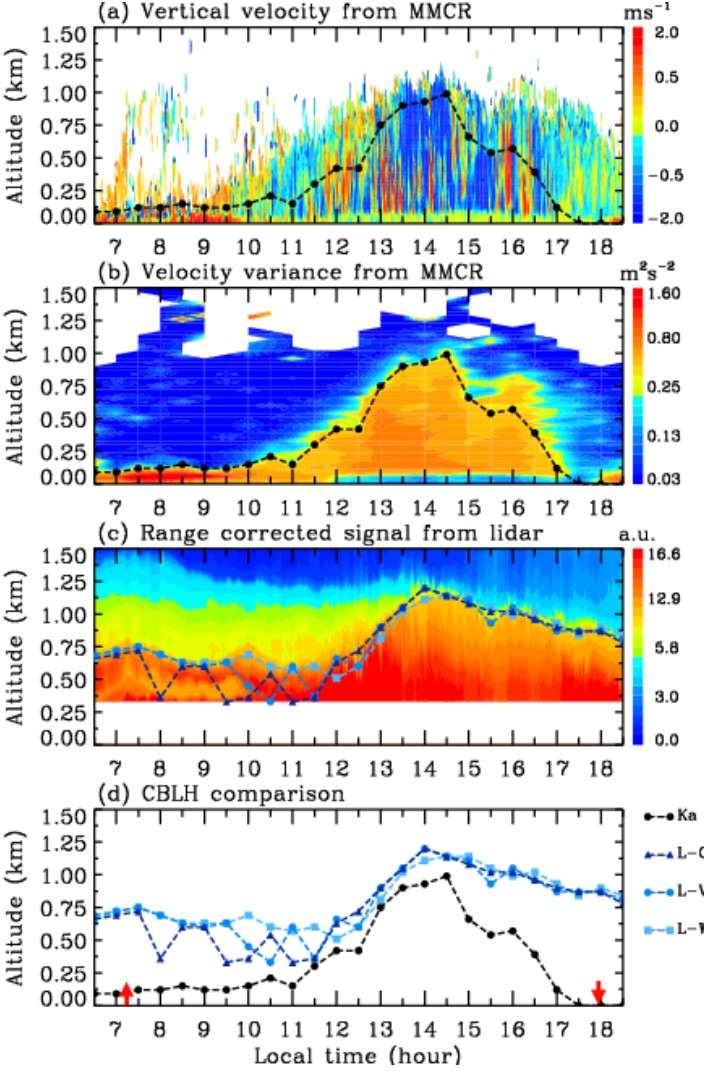

**Figure 6.** Distributions of (a) vertical velocity and (b) its variance from MMCR and (c) lidar RCS on 31

January 2020 with retrieved CBLH, and (d) comparison of CBLHs derived from MMCR and lidar

observations. The threshold of vertical velocity variance from the MMCR is 0.3 $m^2 s^{-2}$. In Panel 6d, the two

red arrows denote the time of sunrise and sunset at 07:15 and 17:57, respectively.



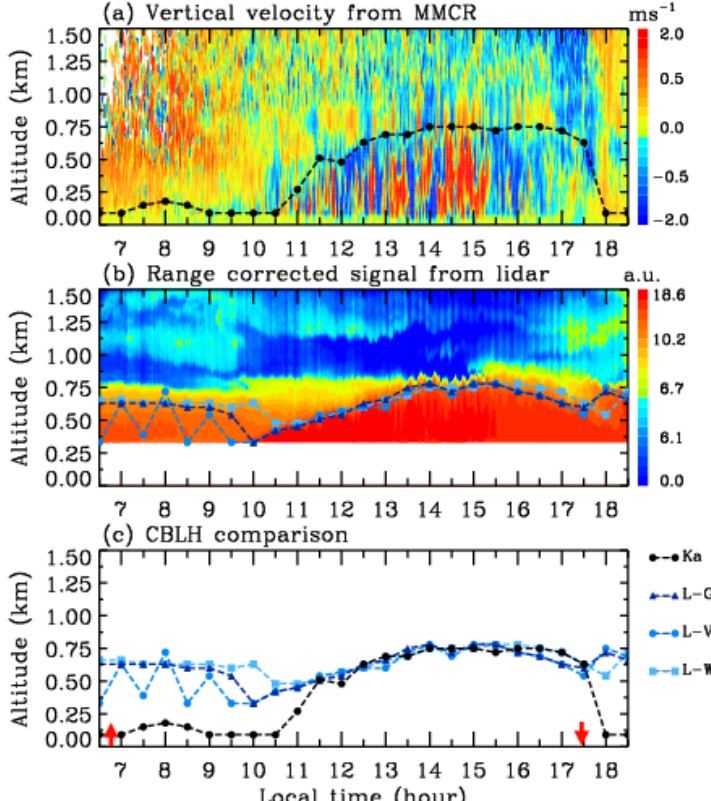

**Figure 7.** Distributions of (a) vertical velocity from MMCR and (b) lidar RCS on 12 November 2020 with

retrieved CBLH, and (c) comparison of CBLHs derived from MMCR and lidar observations. The threshold

of vertical velocity variance from the MMCR is 0.3 $m^2 s^{-2}$. In Panel 7c, the two red arrows denote the time of

sunrise and sunset at 06:47 and 17:27, respectively.

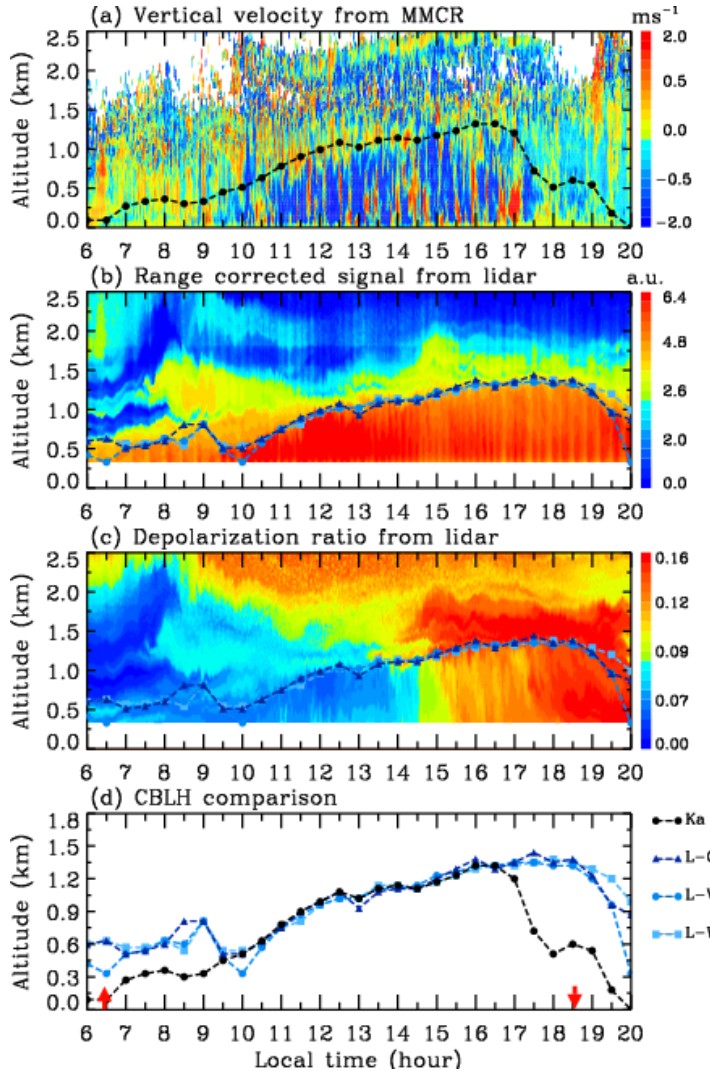


**Figure 8.** Distributions of (a) vertical velocity from MMCR and lidar (b) RCS and (c) depolarization ratio

on 19 March 2020 with retrieved CBLH, and (d) comparison of CBLHs derived from MMCR and lidar
observations. The threshold of vertical velocity variance from the MMCR is 0.3 m$^2$ s$^{-2}$. In Panel 8d, the two
red arrows denote the time of sunrise and sunset at 06:27 and 18:34, respectively.





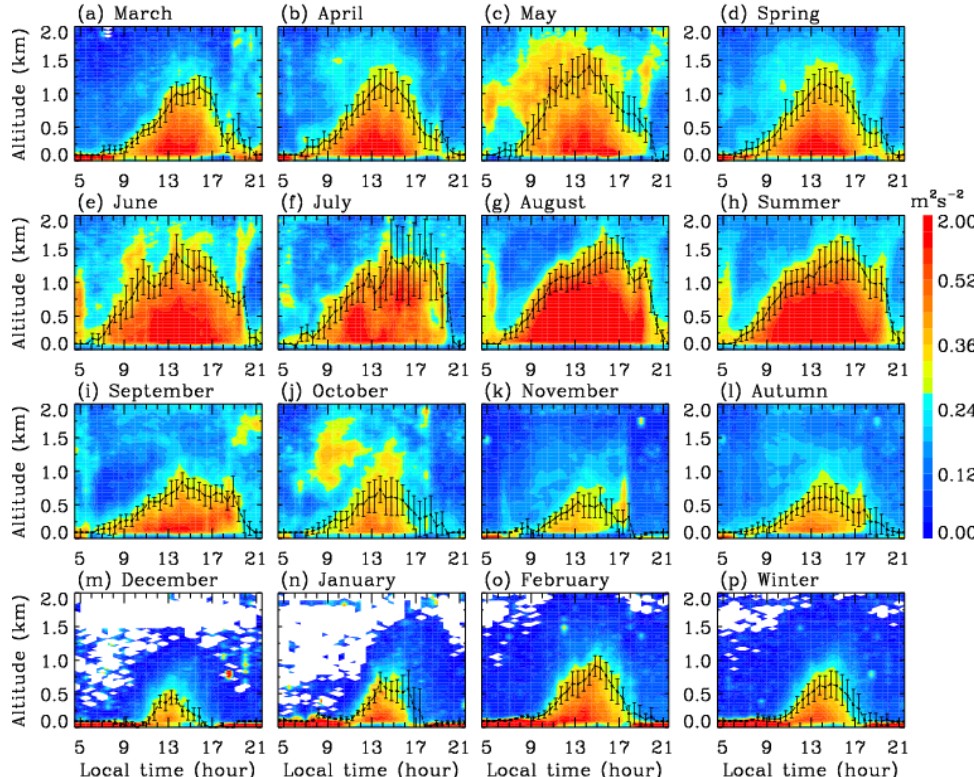

**Figure 9.** Monthly and seasonal mean values and statistical standard deviations of CBLH estimated by threshold of vertical velocity variance from MMCR. The variance threshold is 0.3 $m^2 s^{-2}$, and the color shading denotes the variance distribution. The months and seasons are marked above the corresponding panels, respectively.



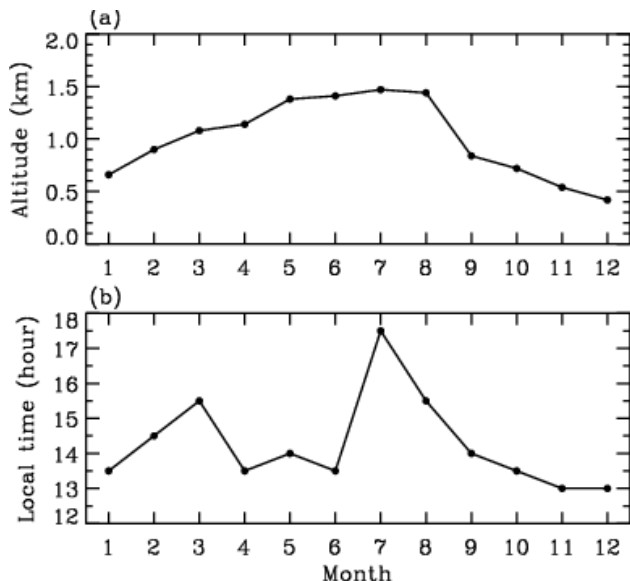

**Figure 10.** (a) Maximum value and (b) occurrence time of monthly mean CBLH derived from threshold of

vertical velocity variance in MMCR observation.



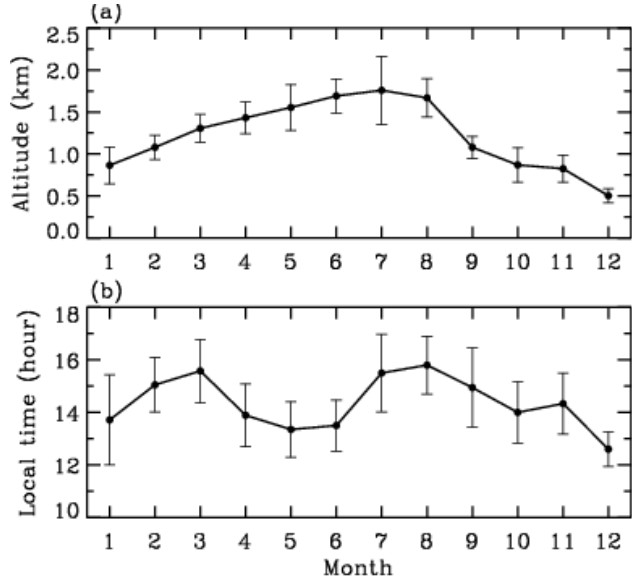

**Figure 11.** Monthly mean values and statistical standard deviations of (a) daily maximum CBLH and (b) its

occurrence time derived from threshold of vertical velocity variance in MMCR observation.