# Peer review of "Technical note: Evolution of convective"

_EGUsphere, 2024_

## Referee Comment (RC1)

**Review on Manuscript " Technical note: Evolution of convective boundary layer height estimated by Ka-band continuous millimeter wave radar at Wuhan in central China"**

**General comments:**

The manuscript offers basically two estimations of Convective Boundary Layer Height (CBLH) for Wuhan, one based on the variance of the vertical velocity from Ka-band cloud radar (MMCR) and one based on the range corrected signal (RCS) from lidar. These two techniques are compared in study-cases and the variance of the vertical velocity from MMCR is also investigated seasonally and monthly. The investigation is relevant and results are interesting, my recommendation would be to eventually accept the present manuscript. However, before it is suitable for publication there are some major revisions that need to be carefully addressed.

1. The focus need to be more precisely explained by the authors since the beginning. Mainly the methodology of RCS and the one of variance of vertical velocity are employed to derive CBLH and compared to each other. Furthermore, the authors investigate some study-cases (during August, November and March 2020) and then the monthly and seasonal variability of CBLH during the year 2020. However, it is not clear why the dates of the study-cases were chosen, that should be clearly stated. Moreover, it is not clear if they want to disentangle the variations of the two methodologies through the seasonal variability or they want to focus on particular dynamics of the study-cases. The focus and novelty of the present research should be much more clear and precise in order to be suitable for publication.

2. There are many discussions that need to be further explained. The physical mechanisms that could be responsible for the difference between the two employed methodologies should be clear. Also, when attributing results to the "history of mixing" or to "seasonality "or "weather conditions", there is often a lot of analysis and specifications lacking. This needs to be better addressed. Specific comments on that are mentioned later in the present review.

3. The introduction includes a lot of basic concepts of boundary layer that need to be better summarized. In its current state, the introduction is quite repetitive and includes too much basic information that is not necessarily relevant for the particular investigation nor referred to in the results.

**Specific comments:**

In the next comments the text in the Manuscript is inside "these marks", while when it is necessary to delete it, it is  and the red text is the suggested added words or phrases.

Line 31: "The seasonal mean CBLH reaches the peak heights of 1.29 km in summer, 1.14 km in spring, and 0.6 km in autumn and winter, with occurrence time between 13:30 and 15:00 LT." For being in the abstract, it would fit better to mention what do higher or lower CBLH mean or put it in terms of mean and standard deviation. What do the authors want to focus in here? Seasonal variability? Diurnal variability? That should be clear.

Line 40: " Atmosphere boundary layer is located in the lowermost layer of the troposphere, and directly impacts the air-land/sea interaction" It is not very precise, authors should better say something like:   "The Atmospheric Boundary Layer (ABL) is located in the lower part of the troposphere, it is directly impacted by the surface forcings and thus it is in the ABL that the land-atmosphere (or land-sea) interaction takes place."

Lines 45 and 46: "Typically, the boundary layer top varies diurnally following the local surface temperature with a magnitude from a few tens of meters to several kilometers." This sentence is not clear, what do authors mean by "following the local surface temperature"? I assume they refer to the surface forcing because of the solar radiation that heats the ground, but that should be explained, because there is no such a thing like "local surface temperature". And then, it is also not clear what has a magnitude from a few tens of meters to several kilometers. I assume they are referring to the ABL height, but that should be explained, as well as the concept of ABL height.

From line 47 to line 54: The evolution of the ABL is explained, but that is very basic knowledge that needs to be better summarized and the focus of what part of this knowledge is useful for the present manuscript should be more clear.

From line 58 to line 66: The authors are again explaining the diurnal evolution of the ABL, so it becomes repetitive and very basic explanations are re-phrased. The whole introduction should be re-structured in order to concisely explain the evolution of the ABL and the different zones that develop within it, and then the observations and methods that are used to estimate CBLH and its variations (there are also very basic concepts about this from line 75 to 80 that need to be better summarized). And the focus on what part of this is relevant to the present manuscript should be clear. In general a much more accurate and concise introduction is required.

From line 120 to line 132: Better write: " At the initial stage of CBL formation in the morning and the rapid decline stage of CBL in the afternoon, aerosol particles in the residual layer may cause the CBLH to be overestimated by about several hundred meters. This discrepancy is due to  often reflects the historyical effect of aerosol mixing rather than the current situation of convectively driven turbulence (Burgos-Cuevas et al., 2023; Schween et al., 2014; Pearson et al., 2010). In the When utilizing Doppler lidar data Doppler lidar observation, a specified threshold of vertical velocity variance is used to define the height of CBL top. This method has been validated as reliable by comparison with the measurements from other equipment, and the sensitivity of threshold has been discussed across different sites (de Arruda et al., 2018; Manninen et al., 2018; Schween et al., 2014; Barlow et al., 2011; Pearson et al., 2010). In contrast to lidar with  A disadvantage of lidar is that it has a large blind range and limited penetrating cloud capability, because of that it is valuable to utilize microwave cloud radar that offers good low altitude coverage and superior performance in cloud penetration. In the cloud observation, there always exist a weak echo layer near the surface, from which the vertical velocity can be retrieved. However, there are few reports utilizing on the use of vertical velocity obtained from Doppler cloud radar for the CBL investigations. "

Line 126: Which other equipment are the authors referring to? Radiosondes?

Line 142: "In this study, the CBLH derived from the MMCR measurements observation is compared..."

Lines 146 to 151: " The climate of the city is humid, dominated by the subtropical monsoon , which is characterized  by abundant precipitation and four distinct seasons (Guo et al., 2023). Due to heavy traffic and industrial activities, large amounts of aerosols are emitted from the industrialized metropolis.  sandstorms from the northwest often pass through Wuhan, especially in spring. These sandstorms cause the remarkable variation in the spatial distribution and concentration of aerosols."

Line 155: Important technical information about the MMCR is missing. What brand is it? What retrieval does it utilize? Is the mode utilized the only possible? What other modes are there and why is this one chosen?

Line 171: What do authors refer to by "plankton"?

Line 174: Remove "and so on" This kind of expressions are not accurate and therefore not suitable for a scientific paper.

Lines 179-200: All section "2.2 Polarization Lidar" needs to be re-structured, information about the location of the lidar and its technical characteristics are merged together. Then authors also talk about how the data is transmitted and retrieved. However, ideas should be more clearly addressed in a more ordered way. Perhaps first talk about location of the lidar, then about the most important lidar measurements characteristics (resolution, brand, configuration...) And finally about how the lidar measures and how it retrieves relevant information. The final paragraph (lines 198-200) is completely out of context, but that information should be mentioned in a more ordered manner.

Lines 203 and 204: "." Not clear, it needs to be completely rewritten in something like: " Given that the CBLH is estimated from instruments that retrieve different variables, the algorithms that are utilized to make such estimations are also based on different principles that are explained in the following subsections.

Line 209: A sentence should be added before start talking about "The wavelet covariance..." This added sentence should brefly explain the physical mechanism because of which aerosol concentration is utilized as a proxy of ABL height. (Because it is assumed that the aerosols are able to mix below the ABL height).

Line 215: Authors need to specify variance of which variable are they referring to.

Line 216: what do authors mean by "temporal domain", they should explain this more precisely.

Lines 216-218: "The frequent exchange of matter and energy between the boundary layer and the free atmosphere causes the dramatical variation of aerosol concentration on small time scales around the CBL top." This sentence needs a reference.

Lines 220 and 221: "We estimate the CBLH from the lidar RCS every 30 min by using the three methods, and then the obtained height is marked at the central time of 30 min." What do authors mean by "marked at central time"? Please explain and re-write.

Line 223: Authors use VV for vertical velocity, it is more usual to utilize letter w for this and the greek letter sigma for the variance.

Line 230: Please add how many measurements does that lapse include.

Line 232: Why is 15th of August 2020 chosen? Any particular interest on this day study case? Are there any statistics to address how the three algorithms compare with each other?

Line 243: "demonstrates that the MMCR VV variance is a fine proxy in the estimation of CBLH" How is this demonstrated if authors are simply comparing particular heights from the three methodologies in a particular study-case? Which one is more reliable? What processes are responsible for the matching or differences of these values? What are the synoptic conditions in the study-case? Is there any sensitivity study regarding that? Is there any idea of how seasonality affects the CBLH retrievals and how to relate this with your study-case?

Lines 246 to 252: Authors discuss here figure 4 in which CBLH is estimated for the same day with different thresholds, so a comparison and analysis of how this height varies is presented. However this is poorly discussed because the threshold is seen to highly impact the estimated CBLH in the morning growing phase and in the afternoon decaying phase of the boundary layer. Authors only mention particular time in the day but a more comprehensive explanation of these phases and the boundary layer evolution and dependence on the thresholds is lacking. As said in lines 247 and 248 the CBLH does not abruptly changes with the thresholds from 9:30 to 17:30, however this corresponds to the developed phase where a fully convective boundary layer is expected. Further analysis on the growing and decaying phases is required, as well as a comparison with other boundary layer height estimation with other thresholds that could also help to argue why the threshold of 0.3 m^2s^-2 is chosen. Also, it would be helpful to include sunrise and sunset times.

Lines 259-261: " It is interesting that the CBLH from the lidar RCS variance drops at 07:30, and then shows a change similar to that from the MMCR VV variance." Authors should mention what physical mechanism could be responsible for that behavior that is coincident with the two techniques.

Lines 264-267: One can note from the reflectivity factor distribution in Figure 5b that cirrus clouds occur from 17:00, develop rapidly into the thick clouds at about 11-14.4 km at 17:30, and then dissipate quickly after 17:30. In the MMCR observation, the cirrus appearance makes a large contribution to a clear dip in the CBLH between 17:30 and 18:30,..." It is not clear what is the interest or the particular feature that the authors want to study with this. Please explain how this findings relate to the focus and in the context of your research.

Lines 269-270: "The influence of clouds on the CBLH is also reported in some earlier studies (Dewani et al., 2023; Bianco et al., 2022; Barlow et al., 2011)." Please explain more precisely how the clouds influence the CBLH.

Line 271: What do authors refer to here when they say "subsidence"? Because subsidence is usually understood as a large scale process that implies synoptic stable conditions and it is not clear what this has to do with the CBLH subsidence. Do authors want to talk about a contraction or a decayment? That is not the same as subsidence.

277: The authors mention the 3 days that were selected, however there is no explanation for making this selection, please include a reason that explains this and hopefully validates that the comparison of different methods for estimating CBLH during these days is relevant.

Line 279: "It is very cold in January at.." In this case "very cold" results rather subjective, so please say how cold or in comparison to what.

283: "Thereafter, the top of CBL  escalates quickly to.."

Lines 287- 289: "This implies that a moderately smaller threshold may be appropriate for the estimation of CBLH in winter with weak turbulence..." Following the discussion before, I don'e see the argument for this implication. Authors need to clarify this. Do they trust more on the RCS? And why? What ABL physical mechanisms are the different methodologies reflecting? I find a lack of discussion here that needs to be better addressed.

Lines 301-302: "In spring, sandstorms occur frequently in the northwest of China, and sand and dust with different intensities are often blown to Wuhan" Is there any particular interest in studying dust storms in spring? Is there any relationship for instance with the amount of dust and the resulting RCS or backscatter that the authors could analyze the impacts? Does that make any difference for the retrievals and their comparison during spring?

Line 316: " lidar measurement  utilizing the VV change in the time domain"

Line 318: "of aerosol residual layer, the CBL tops from the MMCR and lidar  CBLH retrievals are in good agreement with"

Line 321: "Hence, the MMCR VV observation can capture the CBLH evolution very well under a..." Why do authors state that CBLH evolution is "very well" captured? This term sounds subjective and there is a lack of explanation of what do they mean by that.

Line 323: What do authors refer to "seasonal characteristics of convection"? That needs to be clarified. Convection can be related to more larger-scale processes and stable or unstable tropospheric conditions that may or may not include humidity; which usually also implies a seasonality that is not mentioned here. Or it can be more related to radiative driven diurnal cycle convection and then it also can have a seasonality related to variations of radiation through the year, but that is also not explained.  Furthermore on that same line, it is not clear why a "slightly smaller threshold may be more suitable". Please explain why.

Line 331: " We consider that winter covers the months of December, January and February, while March, April and May are spring, June, July and August are summer and the rest is autumn"

Line 333: "As we expect,.."

Line 334: "As the spot of direct sunlight slowly moves northward, the mean variance gradually increases" While looking at the figure, it is clear that not only the intensity of the variance increases but also and more importantly, the height up to which these large values are reached also increases, as well as the time duration of them. These facts should be included in the current analysis.

Line 335: "... August, and then decreases  gradually..."

Line 336: "variance is significantly larger in spring than in autumn" Please be more specific, here you could add some numbers.

Lines 336 and 337: " These monthly and seasonal features of convectively driven turbulence dominate the evolution of monthly and seasonal mean CBLHs." This is a sentence with high repetitiveness and not really with any information, please re-write it or don't include it.

Lines 342-345: Authors analyze figure 10, where maximum value of CBLH is presented for the 12 months. However it is not clear why is this maximum value chosen and there is a lack of analysis on the seasonality that this data implies. Furthermore, figure 10 also shows the local

time when these maximum values were reached but this is not further analyzed nor explained, what does it imply? Then They start a sentence on line 344 saying " In weather forcast record there are 7,3,13,3 and 0 days with moderate  to heavy rain..." But there is a lacking of a connecting sentence that connects this idea with the one in the previous sentence. Why are authors addressing heavy rain here? What do these numbers mean?

Lines 351-357: " As shown in Figure 9, the CBLH in July has the largest standard deviation (between 13:00 and 19:00) and the latest occurrence time of maximum value over the whole year, which is possibly attributable to the cloudy and rainy weather in addition to the strongest radiation. Similarly, the variability of weather conditions may be a major reason why the maximum height arises 1-2 hours earlier in April-June than in March. Nevertheless, with the gradual decline of solar radiation, the occurrence time of maximum height is steadily advanced from 17:30 in July to 13:00 in November and December." This explanation about figure 9 should be before in the Manuscript, before talking about figure 10, please move it and make it consistent.

Lines 363-368: " The occurrence time of averaged maximum CBLH is the earliest at about 12:40 in December and the latest at 15:45 in August, which is slightly distinguished from those in the maximum value of mean CBLH. The standard deviation of occurrence time is obviously large in January, July and September. These results imply that the maximum height and its occurrence time of daily CBL are significantly influenced by the weather conditions besides radiation since the VV variance as a proxy of convectively driven turbulence is sensitive to the weather changes."  These lines first are purely descriptive but I don't see any elucidating analysis on how this maximum CBLH at different times during different months are showing any new information. Again there is a lack of discussion about the processes that can be responsible for those variations. And when the authors mention the "weather conditions" they should be more specific i which conditions are they referring to and how do they change seasonally. Please re-write this making a valuable physical analysis or maybe even don't use the plots on figure 10 if they don't have any conclusive fact about it.

Lines 371-373: " In this study, we investigate the diurnal evolution of monthly and seasonal mean CBLH at Wuhan by the VV variance method based on the Ka-band MMCR observation, and compare the CBLH evolution with that by the RCS gradient, variance and wavelet methods from the lidar measurement." You should add a sentence mentioning that also some study cases were investigated and why those particular study cases; how do they relate to your findings.

Line 374: Be more specific when saying "statistically analyze" what do authors refer to by that?

Lines 379-381: "The occurrence times is between 13:00 in November and December and 17:30 in July for the maximum value of monthly mean CBLH, but between about 12:40 in December and 15:45 in August for the monthly mean value of daily maximum CBLH, respectively." Authors should explain why are they investigating these times and specifically metioning them even in the summary. I don't see any clear elucidating analysis about these times.

Line 382: "  behavior, the  mean CBLH has the maximum heights of 1.29 km at 14:30 and 15:00 in summer, 1.14"

Lines 383-384: " These results are similar to those in earlier studies.." Similar in what sense? The variables, seasons, comparison between methodologies and its particular characteristics or

some more objective facts need to be specified in this sentence. Also maybe it is worth mentioning the influence of different terrain, latitudes, synoptic conditions. How is this comparable?

Lines 391-392: " ...the CBLH from the lidar RCS is higher than from the MMCR VV variance, due to the high blind range of lidar and the strong influence of aerosol residual layer on the lidar RCS." It is not clear why is this due to the high blind range of lidar. Please explain further.

Line 395: " Additionally, in comparison to the lidar RCS affected by the history of aerosol mixing processes, the CBLH" It is not clear how are the authors attributing this to the history of aerosol mixing processes. Please address it more specifically.

Line 407: Please specify what "weather conditions" in particular are the authors referring to. They should include maybe how was temperature, pressure, humidity, was there rain? How were the synoptic conditions on the investigated days. This implies that the authors may need to consider more information and make extra plots showing this or justifying it in a different manner.

---

## Referee Comment (RC2)

**Major comments:**

1.  Too many expressions are used to describe the planetary boundary layer (PBL) in this manuscript, including boundary layer, boundary, CBLH. The PBL can be basically divided into convective boundary layer, neutral boundary layer and stable boundary layer, according the atmospheric static conditions. Therefore, one of my greatest concern is the topic of this work is the evolution of CBL height based on the measurments from MMCR. To my knowledge, the determination of CBL needs the temperature profiles. But I can not see any such profiles in the retrieval process of CBL height.

2.  My second concern lies with the physical basis for the PBL height retrieval from MMCR. As the authors stated in the Introduction section, "there are few reports on the use of vertical velocity obtained from Doppler cloud radar for the CBL investigations." To the best of my knowledge, the cloud-topped PBL is extremely complex due to the complicated turbulence-convection interaction, and the entraiment/detrainment process near the cloud edges. Nevertheless, the MMCR can not efficiently obtain any information (e.g., the vertical velocity) in the absence of cloud, which exactly corresponds to the cloud-topped PBL. Even though the authors say that there exists a weak echo layer near the surface, this could be due to the clutters. If not, the PBL top is well above the near surface layer. Then I pose a question "what is the physical basis for the CBLH and how reliable?"

    If the authors can not well addressed this concern, I recommend rejecting consideration of this work for publication at ACP.

**Specific comments:**

Lines50-51: it is totally wrong to state that "In the afternoon, ***turbulent activity is is weakened". Conversely, in the absence of cloud or synoptic-scale weather system, the turbulence tends to reach the maximum, due to the strongest sensible heat flux in the afternoon.

Lines 51-52: Except for the existence of aerosol particles in the residual layer, most of them are present in the nocturnal stable PBL.

Line 68: What is the "the boundary top"? is it different from the atmospheric boundary layer top? If not, why not use the same term throughouth the whole manuscript?

Lines 76-77: "The radiosonde data have a widely geographical distribution and long-term accumulation" should be rephrased.

Line 78: "boundary layer" refers to "planetary boundary layer height ()"?? if so, what are the difference between PBLH and CBLH?

Line 92-96: Too long sentence for "Wind profile radar can … turbulence (Liu et al., 2020…" and thus I suggest to rephrase it.

Line 109: I do not understand the meaning of "Plunge".

Line 114: "capability" -> "incapability"

Line 121: Is there a rapid decline stage of CBL in the afternoon?? If so, some necessary references are needed to be provided here.

Line 123: Please elaborate on the definition of "historic effect" .

Language: There are too many grammar errors or inappropriate expression throughout the whole manuscript. I can not continue the reviewing processes if the authors did not seek help from a native English editor or colleague.

---

## Author Comment (AC1)

Response to Referee 1

We are deeply grateful to you for your valuable comments and suggestions, which is of great help in improving our manuscript.

According to your suggestions, we revised our manuscript, and presented a point-to-point reply to your comments.

**General comments**

1. The focus need to be more precisely explained by the authors since the beginning. Mainly the methodology of RCS and the one of variance of vertical velocity are employed to derive CBLH and compared to each other. Furthermore, the authors investigate some study-cases (during August, November and March 2020) and then the monthly and seasonal variability of CBLH during the year 2020. However, it is not clear why the dates of the study-cases were chosen, that should be clearly stated. Moreover, it is not clear if they want to disentangle the variations of the two methodologies through the seasonal variability or they want to focus on particular dynamics of the study-cases. The focus and novelty of the present research should be much more clear and precise in order to be suitable for publication.

As you know, the lidar measurement is widely used to study the CBLH, however, this is not the case with the MMCR measurement. The four study-cases demonstrate that the MMCR measurement can obtain the CBLH diurnal evolution in different seasons, especially for the initial growth and final decay phases, meanwhile, the CBLH derived from the MMCR vertical velocity variance is less contamination by passing sand and dust and thick high-level clouds relative to that derived from the lidar RCS. On this basis, we investigate the monthly and seasonal features of CBLH over inland mid-latitudinal Wuhan based on the MMCR observation.

The four cases are not specifically selected, except to exclude precipitation, and severe sand and dust. Severe sand and dust cause the CBLH to be unidentifiable from the lidar RCS, whereas, this scenario is relatively common at Wuhan in spring. According to your suggestion, the chosen case is explained when each case is investigated.

According to your suggestion, we rewrite the last paragraph in Section 1, in which we

further explain the purpose and work in this study. Please see the revised manuscript.

2. There are many discussions that need to be further explained. The physical mechanisms that could be responsible for the difference between the two employed methodologies should be clear. Also, when attributing results to the "history of mixing" or to "seasonality "or "weather conditions", there is often a lot of analysis and specifications lacking. This needs to be better addressed. Specific comments on that are mentioned later in the present review.

In the revision, we add some detailed explanation, for example, the more rapid response of aerosol vertical velocity to thick high-level clouds than its concentration. Please see more explanation in the Response to Comments later.

3. The introduction includes a lot of basic concepts of boundary layer that need to be better summarized. In its current state, the introduction is quite repetitive and includes too much basic information that is not necessarily relevant for the particular investigation nor referred to in the results.

According to your constructive suggestion, only PBL and CBL are presented in the revised manuscript, and the repetitive basic information is rewritten or deleted. Thus, the introduction has been simplified, and please see the revision and the Response to Comments later.

**Specific comments:**

1. Line 31: "The seasonal mean CBLH reaches the peak heights of 1.29 km in summer, 1.14 km in spring, and 0.6 km in autumn and winter, with occurrence time between 13:30 and 15:00 LT." For being in the abstract, it would fit better to mention what do higher or lower CBLH mean or put it in terms of mean and standard deviation. What do the authors want to focus in here? Seasonal variability? Diurnal variability? That should be clear.

According to your suggestion, the sentence is changed as,

"The MMCR observation in 2020 depicts the diurnal evolution of seasonal and monthly mean CBLHs. The seasonal mean CBLH reaches the peak heights of 1.29 km in summer,

1.14 km in spring, 0.66 km in autumn, and 0.6 km in winter, indicating the dominant effect of radiation heating."

2. Line 40: " Atmosphere boundary layer is located in the lowermost layer of the troposphere, and directly impacts the air-land/sea interaction" It is not very precise, authors should better say something like: "The Atmospheric Boundary Layer (ABL) is located in the lower part of the troposphere, it is directly impacted by the surface forcings and thus it is in the ABL that the land-atmosphere (or land-sea) interaction takes place."

According to your suggestion, we reword the sentence to be,

"The planetary boundary layer (PBL) is located in the lower part of the troposphere, and is where the air-land (or air-sea) interaction takes place, thus the PBL is directly impacted by the surface forcings."

3. Lines 45 and 46: "Typically, the boundary layer top varies diurnally following the local surface temperature with a magnitude from a few tens of meters to several kilometers." This sentence is not clear, what do authors mean by "following the local surface temperature"? I assume they refer to the surface forcing because of the solar radiation that heats the ground, but that should be explained, because there is no such a thing like "local surface temperature". And then, it is also not clear what has a magnitude from a few tens of meters to several kilometers. I assume they are referring to the ABL height, but that should be explained, as well as the concept of ABL height.

Considering the dfferent driving mechanisms and influencing factors of PBL, the sentence is corrected in the revision as,

"The height of PBL varies with local time, ranging generally from a few tens of meters to several kilometers at mid latitudes".

4. From line 47 to line 54: The evolution of the ABL is explained, but that is very basic knowledge that needs to be better summarized and the focus of what part of this knowledge is useful for the present manuscript should be more clear.

According to your suggestion, the diurnal evolution of CBL is transferred to the second paragraph and is reduced to be,

"On a clear day, the CBLH rises after sunrise and reaches its maximum in the afternoon."

5. From line 58 to line 66: The authors are again explaining the diurnal evolution of the ABL, so it becomes repetitive and very basic explanations are re-phrased. The whole introduction should be re-structured in order to concisely explain the evolution of the ABL and the different zones that develop within it, and then the observations and methods that are used to estimate CBLH and its variations (there are also very basic concepts about this from line 75 to 80 that need to be better summarized). And the focus on what part of this is relevant to the present manuscript should be clear. In general a much more accurate and concise introduction is required.

According to your helpful suggestion, we restructure the first and second paragraphs. The first paragraph briefly introduces the PBL, and the second paragraph describes the CBL that we investigate in the manuscript.

We concisely explain the evolution of CBL, and slightly add the influence of cloud and humidity on the CBL, which is relevant to our manuscript. Please see the revised manuscript.

The description of radiosonde in lines 75-80 is briefly rewritten as,

"Radiosonde can obtain many meteorological parameters with high precision, providing the possibility of retrieving the CBLH through different algorithms."

6. From line 120 to line 132: Better write: "At the initial stage of CBL formation in the morning and the rapid decline stage of CBL in the afternoon, aerosol particles in the residual layer may cause the CBLH to be overestimated by about several hundred meters. This discrepancy is due to often reflects the historyical effect of aerosol mixing rather than the current situation of convectively driven turbulence (Burgos-Cuevas et al., 2023; Schween et al., 2014; Pearson et al., 2010). In the When utilizing Doppler lidar data Doppler lidar observation, a specified threshold of vertical velocity variance is used to define the height of CBL top. This method has

been validated as reliable by comparison with the measurements from other equipment, and the sensitivity of threshold has been discussed across different sites (de Arruda et al., 2018; Manninen et al., 2018; Schween et al., 2014; Barlow et al., 2011; Pearson et al., 2010). In contrast to lidar with A disadvantage of lidar is that it has a large blind range and limited penetrating cloud capability, because of that it is valuable to utilize microwave cloud radar that offers good low altitude coverage and superior performance in cloud penetration. In the cloud observation, there always exist a weak echo layer near the surface, from which the vertical velocity can be retrieved. However, there are few reports utilizing on the use of vertical velocity obtained from Doppler cloud radar for the CBL investigations."

Combing the other Referee's suggestion, the paragraph is rewritten in the revision, as follows,

"At the initial stage of CBL formation in the morning and the rapid decline stage of CBL in the late afternoon (Dewani et al., 2023; Manninen et al., 2018; Schween et al., 2014; Barlow et al., 2011), aerosol particles in the residual layer may cause the CBLH to be overestimated several hundred meters. This discrepancy is due to aerosols from a long time-mixing process rather than the current situation of convectively driven turbulence (Burgos-Cuevas et al., 2023; Schween et al., 2014; Pearson et al., 2010). When utilizing Doppler lidar data, a specified threshold of vertical velocity variance is used to define the height of CBL top. This method has been validated by comparison with the radiosonde observation (Dang et al., 2019; Li et al. 2017; Granados-Muñoz et al., 2012), and the sensitivity of threshold has been discussed across different sites (de Arruda et al., 2018; Manninen et al., 2018; Schween et al., 2014; Barlow et al., 2011; Pearson et al., 2010). A disadvantage of lidar is that it has a large blind range and incapability to penetrate clouds, thus because of that, it is valuable to utilize microwave cloud radar that offers good low altitude coverage and superior performance in cloud penetration. In the cloud observation, a weak echo layer always exists near the surface, from which the vertical velocity can be retrieved. However, there are few reports utilizing vertical velocity obtained from Doppler cloud radar for the CBL investigations."

 Line 126: Which other equipment are the authors referring to? Radiosondes?

Yes, as you said.

"with the measurements from other equipment" is changed as "with the radiosonde observation (Dang et al., 2019; Li et al. 2017; Granados-Muñoz et al., 2012)".

8. Line 142: "In this study, the CBLH derived from the MMCR measurements  is compared..."

"observation" is changed as "measurements".

9. Lines 146 to 151: "The climate of the city is humid, dominated by the subtropical monsoon , which is characterized  by abundant precipitation and four distinct seasons (Guo et al., 2023). Due to heavy traffic and industrial activities, large amounts of aerosols are emitted from the industrialized metropolis.  sandstorms from the northwest often pass through Wuhan, especially in spring. These sandstorms cause the remarkable variation in the spatial distribution and concentration of aerosols."

These errors have been corrected in the revision.

10. Line 155: Important technical information about the MMCR is missing. What brand is it? What retrieval does it utilize? Is the mode utilized the only possible? What other modes are there and why is this one chosen?

Conventional microwave radar is pulse radar. After the antenna emits a pulse signal, echoes are received by the same antenna. The frequency-modulated continuous wave radar in the manuscript is a kind of new system radar. The transmitting antenna continuously emits microwaves, thus echoes need to be received by another antenna. The distance information of echoes is retrieved by demodulating the modulated continuous wave. Due to the continuous transmitting and receiving, the continuous wave radar has much larger mean power relative to pulse radar, which is why the MMCR in the manuscript can always receive weak echoes from aerosols near the ground.

According to your suggestion, we add the brand as "WHU-CW MMCR".

We add and rephrase some descriptions, as follows,

"The radar system transmits a mean power of 50 W at operating frequency of 35.035 GHz through 0.38° width beam formed by a Cassegrain antenna with 1.5 m diameter. Backscatter echoes from aerosol and cloud particles are received by the other same Cassegrain antenna";

"Because of almost continuous transmitting and receiving, FMCW radar has generally a much higher mean power relative to pulse radar, which improves the capacity of MMCR to detect weak echo targets. Meanwhile, by modulating and demodulating the continuous wave, the FMCW radar measurement has an adjustable range and time resolution."

11. Line 171: What do authors refer to by "plankton"?

"plankton" is present as "aerial plankton" in early literature, which suggested the weak echoes near the surface from "small insects and aerial plankton". Thus, "plankton and insects" is changed as "small insects and aerial plankton" in the revised manuscript.

12. Line 174: Remove "and so on" This kind of expressions are not accurate and therefore not suitable for a scientific paper.

"And so on" has been deleted.

13. Lines 179-200: All section "2.2 Polarization Lidar" needs to be re-structured, information about the location of the lidar and its technical characteristics are merged together. Then authors also talk about how the data is transmitted and retrieved. However, ideas should be more clearly addressed in a more ordered way. Perhaps first talk about location of the lidar, then about the most important lidar measurements characteristics (resolution, brand, configuration...) And finally about how the lidar measures and how it retrieves relevant information. The final paragraph (lines 198-200) is completely out of context, but that information should be mentioned in a more ordered manner.

According to your suggestion, we restructure the section, as follows,

"The WHU-PL polarization lidar developed by ARSO is also located in Wuhan University, about 0.5 km from the Ka-band MMCR. The lidar telescope is 70 m above sea level, which is about 30 m higher than the MMCR antenna. Expanded laser beam overlaps with the full view field of the receiving telescope at a height of 0.3 km, thus this height is the low limit of lidar detection. The lidar data has a temporal resolution of 1 min, and the same vertical resolution of 30 m as the MMCR data. In this study, we regard the height of MMCR antenna as a baseline, and then the initial height of lidar data is set at 0.33 km.

The lidar system consists of transmitting subsystem receiving subsystem and information processing subsystem. The lidar vertically emits the laser pulses of 120 mJ at operating wavelength of 532 nm with a repetition rate of 20 Hz by a frequency-doubled Nd: YAG laser. The output polarized laser beam has a fine polarization purity with depolarization ratio less than 1:10000 by using a Brewster polarizer. Light backscattered by aerosol and cloud particles and atmospheric molecules is collected by a telescope with 0.3 m diameter. After separated through an interference filter with 0.3 nm bandwidth centered at 532 nm, the elastically backscattered light is incident on a polarization beam splitter prism, and then the two-channel polarized light are focused onto two photomultiplier tubes (PMTs), respectively. The signals from the two PMTs are transferred to a personal computer (PC)-controlled two-channel transient digitizer, which can obtain the echo signal intensity and volume depolarization ratio through the PC processing. Backscatter coefficient is retrieved based on the backward iteration algorithm under the condition of a given lidar ratio proposed by Fernald and Klett (Fernald, 1984; Klett, 1981), and then the RCS is derived from the backscatter coefficient (Freudenthaler et al., 2009; Immler and Schrems, 2003). The lidar configuration and depolarization comparison with the measurement from the cloud-aerosol lidar and infrared pathfinder satellite observation (CALIPSO) were in detail described in the early study (Kong and Yi, 2015)."

14. Lines 203 and 204: "

" Not clear, it needs to be completely rewritten in something like: "Given that the CBLH is estimated from instruments that retrieve different variables, the algorithms that are utilized to make such estimations are also based on different principles that are explained in the following subsections".

In the revised manuscript, the sentence was modified as "Given that the CBLH is estimated from instruments that retrieve different variables, the algorithms that are utilized to make such estimations are also based on different principles, which are explained in the following subsections."

15. Line 209: A sentence should be added before start talking about "The wavelet covariance..." This added sentence should brefly explain the physical mechanism because of which aerosol concentration is utilized as a proxy of ABL height. (Because it is assumed that the aerosols are able to mix below the ABL height).

According to your suggestion, the description before "The wavelet covariance..." is rephrases as,

"In the lidar observation, the CBLH is derived from the RCS, which is approximately proportional to the aerosol concentration (Kong and Yi, 2015; Lewis et al., 2013; Pal et al., 2010; Emeis et al., 2008). Generally, aerosols are well-mixed within the CBL due to the convectively driven turbulence, and its concentration decays sharply over the CBL top. Hence, the gradient (Grd) method is often utilized to investigate the CBLH by identifying the strongest or minimum gradient of RCS."

16. Line 215: Authors need to specify variance of which variable are they referring to.

According to the Comments 16-19, we rewrite the paragraph, as follows,

"On the other hand, because of the entrainment process, there is a frequent exchange of matter and energy between the CBL and the free atmosphere, causing the dramatic variation of aerosol concentration or lidar RCS on small time scales around the CBL top (Zhang et al., 2018; Kong and Yi, 2015). In this case, the variance (Var) method is used

to determine the CBL top by identifying the maximum variance of RCS during a relatively long period (Lammert and Bösenberg, 2006; Martucci et al., 2004; Piironen and Eloranta, 1995). We estimate the CBLH from the lidar RCS in a period of 30 min by using the three methods, for instance, the CBLH at 12:00 LT (the same below) is calculated based on the RCS data from 11:45 to 12:15."

17. Line 216: what do authors mean by "temporal domain", they should explain this more precisely.

The unclear description is removed in the revision, and please see the Response to the Comment 16.

18. Lines 216-218: "The frequent exchange of matter and energy between the boundary layer and the free atmosphere causes the dramatical variation of aerosol concentration on small time scales around the CBL top." This sentence needs a reference.

The reference is added. and please see the Response to the Comment 16.

19. Lines 220 and 221: "We estimate the CBLH from the lidar RCS every 30 min by using the three methods, and then the obtained height is marked at the central time of 30 min." What do authors mean by "marked at central time"? Please explain and re-write.

The sentence is rephrased, and please see the Response to the Comment 16.

20. Line 223: Authors use VV for vertical velocity, it is more usual to utilize letter w for this and the greek letter sigma for the variance.

According to your suggestion, "VV" for vertical velocity is replaced by $w$, and "VV variance" is replaced by $\sigma_w^2$.

21. Line 230: Please add how many measurements does that lapse include.

This sentence has been changed as "the threshold method is also used to determine a CBLH from the more than 6000 $w$ profiles in the MMCR measurement during a

period of 30 min."

22. Line 232: Why is 15th of August 2020 chosen? Any particular interest on this day study case? Are there any statistics to address how the three algorithms compare with each other?

Relative to the MMCR, the lidar has much less working days, especially for all-day operation. 15 August 2020 is chosen because 1) both the radar and lidar have the data; 2) it rained on 12 August 2020, which should have little influence on the analysis of CBLH on 15 due to two days of evaporation; and thus 3) the sunny 15 August with clouds only in the late afternoon have a good evolution of CBL.

By using the available lidar data, we compare CBLHs from the lidar RCS by the three methods and calculate their correlations. Figure S1 (only presented in the Response) shows the comparison of CBLHs between 7:00 and 17:00 from the perspective of local time and season, respectively. The chosen 7:00 and 17:00 are based on the sunrise and sunset in winter. Figure S2 shows the comparison of CBLHs from 2 h after sunrise to 2 h before sunset.

[Figure]

**Figure S1.** Comparison of lidar RCS CBLHs between 7:00 and 17:00 from the perspective of (upper row) local time and (lower row) season. L-G, L-W and L-V represent the Grd, WCT and Var methods, respectively.

[Figure]

**Figure S2.** Comparison of lidar RCS CBLHs from 2 h after sunrise to 2 h before sunset according to the perspective of (upper row) local time and (lower row) season.

As shown in Figures S1 and S2, all the correlation coefficients are larger than 0.8, whereas, there are some differences between them, especially in the in the initial ascent phase. Their differences may be understandable because the three methods have the different physical principles or mathematical techniques.

In the revision, we add the description of "The day is 3 days later than the rainy day of 12 August."

23. Line 243: "demonstrates that the MMCR VV variance is a fine proxy in the estimation of CBLH" How is this demonstrated if authors are simply comparing particular heights from the three methodologies in a particular study-case? Which one is more reliable? What processes are responsible for the matching or differences of these values? What are the synoptic conditions in the study-case? Is there any sensitivity study regarding that? Is there any idea of how seasonality affects the CBLH retrievals and how to relate this with your study-case?

Similar to the comparison of lidar CBLHs in the Response to the Comment 22, we further compare the MMCR CBLH with the lidar CBLHs by using the MMCR data in the same days as the lidar data. The comparison is presented in Figures S3 and S4 in the

form of scatter diagram.

Firstly, all the correlation coefficients between the MMCR CBLH and the lidar CBLHs is more than 0.66, however, these correlation coefficients are generally smaller than those between the lidar CBLHs.

Secondly, the differences between the MMCR and lidar CBLHs occur primarily in the initial ascent phase and the final descent phase, which is consistent with the results in the manuscript. The lower correlation in Figures S3 and S4 than in Figures S1 and S2 can be caused at least by 1) the large blind range of lidar; 2) the influence of aerosol residual layer on the lidar RCS; and 3) the response of aerosol concentration to weather condition variation is slower than the vertical velocity, as shown in the manuscript.

Thirdly, from the perspective of the season, 1) the lidar CBLHs in winter are generally higher than the MMCR CBLH, similar to the case in winter from the manuscript. We agree with your Comments 32 and 37. Since the CBLHs identified from the MMCR and lidar have different physical principles, the weak solar radiation and wet ground in winter may be responsible for the difference of CBLHs in winter. 2) the scatter diagram shows that there are more discrete points in spring relative to the other seasons. There are the frequent sand and dust activities at Wuhan in spring, which make a major contribution to this discrepancy between the MMCR and lidar CBLHs in spring. Additionally, the four cases in the different seasons from the manuscript show the discrepancies between the MMCR and lidar CBLHs, which are generally consistent with the statistical seasonal discrepancies, respectively. Thus, there aren't many causes of careful selecting for the four cases in the manuscript except for excluding the days with rainfall or very severe dust storms.

Finally, in Figures S4 and S4, the correlation coefficient of the MMCR CBLH with the lidar CBLH from Var method is larger than that with the lidar CBLH from Grd and WCT methods, similar to the result in the manuscript. The RCS gradient is the vertically variational rate of 30 min-mean RCS. However, although $w$ and RCS are different physical variables, both their variances represent the deviation degree of their small time scale values relative to their 30 min-mean values, which may be the cause of the higher

correlation between the CBLHs from the two Var methods.

As shown in Figures S1-S4, when the CBL is well developed, the CBLHs derived from different methods have a higher consistency. As you implied, a particular case in the noon is not enough to draw a conclusion, hence, we delete "Hence, the good consistency of CBLH derived from the MMCR and lidar demonstrates that the MMCR VV variance is a fine proxy in the estimation of CBLH." in the revised manuscript.

[Figure]

**Figure S3.** Comparison of MMCR CBLH with lidar RCS CBLHs between 7:00 and 17:00 from the perspective of (upper row) local time and (lower row) season. Ka represents the Var method from Ka-band MMCR, and L-G, L-W and L-V represent the Grd, WCT and Var methods from lidar, respectively.

[Figure]

**Figure S4.** Comparison of MMCR CBLH with lidar RCS CBLHs from 2 h after sunrise to 2 h before sunset according to the perspective of (upper row) local time and (lower row) season.

24. Lines 246 to 252: Authors discuss here figure 4 in which CBLH is estimated for the same day with different thresholds, so a comparison and analysis of how this height varies is presented. However this is poorly discussed because the threshold is seen to highly impact the estimated CBLH in the morning growing phase and in the afternoon decaying phase of the boundary layer. Authors only mention particular time in the day but a more comprehensive explanation of these phases and the boundary layer evolution and dependence on the thresholds is lacking. As said in lines 247 and 248 the CBLH does not abruptly changes with the thresholds from 9:30 to 17:30, however this corresponds to the developed phase where a fully convective boundary layer is expected. Further analysis on the growing and decaying phases is required, as well as a comparison with other boundary layer height estimation with other thresholds that could also help to argue why the threshold of 0.3 m^2s^-2 is chosen. Also, it would be helpful to include sunrise and sunset times.

According to your constructive suggestion, we compare the MMCR CBLH with the radiosonde CBLH. The radiosonde data are provided by University of Wyoming at the website of https://weather.uwyo.edu/upperair/bufrraob.shtml. The Radiosonde data at

Wuhan have a very rough vertical resolution of about 0.5-1.0 km before June 2021, and since then, the vertical resolution has been improved to tens to hundreds of meters in the thousands of meters above the surface. The sun has gone down at 20:00, thus we present the comparison at 8:00 in some days without precipitation.

We identify the CBLH by the maximum gradient of potential temperature ($\theta$) and a 0.25 threshold of bulk Richardson number ($Ri$) based on the radiosonde data (Guo et al., 2021; Seidel et al., 2010; Seibert et al., 2000). The bulk Richardson number is expressed (Guo et al., 2021; Seidel et al., 2010; Seibert et al., 2000) as follows:

$$Ri(z) = \frac{\left(\frac{g}{\theta_{vs}}\right)(\theta_{vz} - \theta_{vs})z}{(u_z - u_s)^2 + (v_z - v_s)^2 + (bu_*^2)} \,,$$

where $g$ is the gravitational acceleration; $z$ is the height; $\theta_v$ is the virtual potential temperature; $u_*$ is the surface friction velocity; $u$ and $v$ is the zonal and meridional wind components, respectively; and $b$ is a constant, which is usually set to zero due to the fact that friction velocity is much weaker compared with the horizontal wind (Seidel et al., 2012). The subscripts of $z$ and $s$ denote the parameters at $z$ height and surface level, respectively.

Figure S5 presents the comparison of CBLHs derived from the three methods at 8:00 in some days. Firstly, as shown in Figure S5, the CBLH from the threshold of $\sigma_w^2$=0.3 m$^2$ s$^{-2}$ from MMCR is generally consistent with the CBLHs from the gradient of $\theta$ and the threshold of $Ri$=0.25 from radiosonde, which indicates that the threshold of $\sigma_w^2$ =0.3 m$^2$ s$^{-2}$ should be appropriate for the CBLH estimation at Wuhan. Nevertheless, there is also a little difference in the results from the three methods because theses methods have the different physical principles.

Secondly, the CBLH at 8:00 shows a change from day to day in the same month, and even in three straight sunny days of 20-22 July. A possible cause is that the other factors may have a relatively important impact on the CBLH in the initial growing phase relative to the turbulence driven primarily by surface heating in the noon.

Thirdly, when there is a rapid reduction around $\sigma_w^2$=0.3 m$^2$ s$^{-2}$ with height increasing,

the estimated CHLH is insensitive to the thresholds. Whereas, in the initial growing phase, the slow reduction around $\sigma_w^2$=0.3 m² s⁻² may occur, too, leading to that the threshold is seen to highly impact the estimated CBLH. We conjecture that the relatively important impact of the other factors on $\sigma_w^2$ besides surface heating may be responsible for the $\sigma_w^2$ variation in the initial and final phases.

Sorry, this MMCR did not work routinely for many days or several months in 2021 due to the epidemic and the transmitter returned to the factory for maintenance, thus the MMCR data in only 2020 is used in the investigation.

According to your suggestion, we add the sunrise and sunset times in Figure 4, and the sentence as follows,

"By comparison (not presented), the CBLH at 8:00 derived from the threshold of $\sigma_w^2$ =0.3 m² s⁻² is generally consistent with the CBLHs estimated by the maximum gradient of potential temperature and a 0.25 threshold of bulk Richardson number based on the radiosonde data at Wuhan, which indicates that the threshold of $\sigma_w^2$=0.3 m² s⁻² is appropriate for the CBLH estimation at Wuhan".

[Figure]

**Figure S5.** Comparison of CBLHs estimated by the $\sigma_w^2 =0.3\ \mathrm{m^2\,s^{-2}}$ threshold from MMCR, and the maximum gradient of $\theta$ and $Ri$=0.25 threshold from radiosonde data at 8:00. There are the three panels in each day, and the blue horizontal line denotes the position of estimated CBL top. In the panel (a), the gray thin and black thick lines denote $w$ and their mean values from MMCR, respectively; the red line denote $\sigma_w^2$; and the yellow line denotes the threshold of $\sigma_w^2$ =0.3 $\mathrm{m^2\,s^{-2}}$. In the panel (b), the black and red lines denote $\theta$ and its gradient from radiosonde, respectively. In the panel (c), the black line denotes $Ri=$ from radiosonde data, and the yellow line denotes the threshold of $Ri$=0.25.

25. Lines 259-261: "It is interesting that the CBLH from the lidar RCS variance drops at 07:30, and then shows a change similar to that from the MMCR VV variance." Authors should mention what physical mechanism could be responsible for that behavior that is coincident with the two techniques.

As shown in Figures S3 and S4, the statistical results also indicate that the correlation coefficient of the MMCR CBLH with the lidar CBLH from Var method is larger than that with the lidar CBLH from Grd and WCT methods.

In the revised manuscript, we add the explanation as,

"Both the variances of $w$ and RCS represent the deviation degree of their small time scale values relative to their 30 min-mean values, which may be responsible for the similar results."

26. Lines 264-267: One can note from the reflectivity factor distribution in Figure 5b that cirrus clouds occur from 17:00, develop rapidly into the thick clouds at about 11-14.4 km at 17:30, and then dissipate quickly after 17:30. In the MMCR observation, the cirrus appearance makes a large contribution to a clear dip in the CBLH between 17:30 and 18:30,..." It is not clear what is the interest or the particular feature that the authors want to study with this. Please explain how this findings relate to the focus and in the context of your research.

Combining the next Comment 27, we understand what you mean, clouds have a complex influence of on turbulence in the CBL, which may cause an insufficiently focused purpose in the study.

The CBLH in our manuscript has an obvious dip in the period, hence, we need to give an explanation that the cloud appearance is responsible for the CBLH dip, and please the response to the next Comment.

27. Lines 269-270: "The influence of clouds on the CBLH is also reported in some earlier studies (Dewani et al., 2023; Bianco et al., 2022; Barlow et al., 2011)." Please explain more precisely how the clouds influence the CBLH.

Early studies from the Doppler lidar measurements reported the influence of low-level clouds ("Higher level clouds not detected by the lidar" in Barlow et al., 2011, and similarly, "This study considers the low clouds" in Dewani et al. 2023) on the CBL and turbulence.

In the revised manuscript, we rewrite the description as,

"In the MMCR observation, the CBLH shows a clear dip between 17:30 and 18:30, and then a lift as the clouds dissipates rapidly. Earlier studies from the Doppler lidar $w$ investigated the complex influence of low-level clouds on the CBL and turbulence. The cloud-top radiative cooling drives top-down convective mixing, leading to the enhancement of $\sigma_w^2$ (Hogan et al., 2009; Harvey et al., 2013; Manninen et al., 2018).

Whereas, during the warm season, the magnitude of $\sigma_w^2$ from the lidar $w$ is large on clear-sky days and decreases on cloud-topped days, and the intensity of turbulence reduces with an increase in the cloud fraction within the CBL, except in the cloud layer that exceeds 90% of the CBL thickness (Dewani et al., 2023). Here, the cirrus clouds are above 11 km, thus the cloud-top driven convective mixing has little impact on the low atmosphere, however, the thick clouds cool the surface by attenuating solar radiation, which can weaken the surface-driven convective mixing. Therefore, the thick cirrus makes a large contribution to the CBLH dip."

28. Line 271: What do authors refer to here when they say "subsidence"? Because subsidence is usually understood as a large scale process that implies synoptic stable conditions and it is not clear what this has to do with the CBLH subsidence. Do authors want to talk about a contraction or a decayment? That is not the same as subsidence.

Sorry for the unclear expression. "subsidence" is replaced with "dip".

29. 277: The authors mention the 3 days that were selected, however there is no explanation for making this selection, please include a reason that explains this and hopefully validates that the comparison of different methods for estimating CBLH during these days is relevant.

We hope to use the chosen four cases for investigating the CBLH evolution in four seasons, and also for confirming these methods.

In the revision, we have added the sentence as

"The three days, without clouds and precipitation, are chosen as the representative in different seasons."

30. Line 279: "It is very cold in January at.." In this case "very cold" results rather subjective, so please say how cold or in comparison to what.

> This sentence is rephrased as "January is the coldest month of the year, and on 31 January, the minimum (maximum) temperature is -5 °C (4 °C) recorded in the weather forecast."

31. 283: "Thereafter, the top of CBL  escalates quickly to.."

> "climbs" is corrected to be "escalates".

32. Lines 287- 289: "This implies that a moderately smaller threshold may be appropriate for the estimation of CBLH in winter with weak turbulence..." Following the discussion before, I don't see the argument for this implication. Authors need to clarify this. Do they trust more on the RCS? And why? What ABL physical mechanisms are the different methodologies reflecting? I find a lack of discussion here that needs to be better addressed.

> As shown in the Response to the Comment 24, the threshold of $\sigma_w^2 = 0.3$ m$^2$ s$^{-2}$ is appropriate for the CBLH estimation at Wuhan. We rewrite the sentence as follows, "it can be seen from Figure 6d that all the CBLHs from the lidar RCS are slightly larger than those from the MMCR $\sigma_w^2$, which may be attributed to the long time-mixing aerosols and wet surface in winter."

33. Lines 301-302: "In spring, sandstorms occur frequently in the northwest of China, and sand and dust with different intensities are often blown to Wuhan" Is there any particular interest in studying dust storms in spring? Is there any relationship for instance with the amount of dust and the resulting RCS or backscatter that the authors could analyze the impacts? Does that make any difference for the retrievals and their comparison during spring?

> We are not particular interested in the study of dust storms in spring. However, the sand and dust are very frequent in spring, thus there are almost always strong or weak (and single or multiple layers) sand and dust in the lidar data in spring. The lidar observation is sensitive to sand and dust, which have a large influence of CBLH estimation from the

lidar RCS, as shown in the statistic results in Figures S3 and S4.

Hence, when we compare the spring results of MMCR and lidar, we cannot explain their differences if sand and dust is not mentioned.

In the revised manuscript, "In spring, sandstorms occur frequently in the northwest of China, and sand and dust with different intensities are often blown to Wuhan." is rewritten to be,

"In spring, sand and dust with different intensities from the northwest of China pass frequently through Wuhan."

34. Line 316: "lidar measurement  utilizing the VV change in the time domain"

The word is changed in the revised manuscript.

35. Line 318: "of aerosol residual layer, the CBL tops from the MMCR and lidar  CBLH retrievals are in good agreement with"

The sentence is rephrased as "the MMCR CBLHs are generally in agreement with the lidar CBLHs."

36. Line 321: "Hence, the MMCR VV observation can capture the CBLH evolution very well under a..." Why do authors state that CBLH evolution is "very well" captured? This term sounds subjective and there is a lack of explanation of what do they mean by that.

This sentence has been rewritten as "Hence, the MMCR observation can accurately retrieve the CBLH and capture its diurnal evolution, especially for the CBL in the blind range of lidar."

37. Line 323: What do authors refer to "seasonal characteristics of convection"? That needs to be clarified. Convection can be related to more larger-scale processes and stable or unstable tropospheric conditions that may or may not include humidity; which usually also implies a seasonality that is not mentioned here. Or it can be more related to radiative driven diurnal cycle convection and then it also can have a seasonality related to variations of radiation through the

year, but that is also not explained. Furthermore on that same line, it is not clear why a "slightly smaller threshold may be more suitable". Please explain why.

Considering that the monthly and seasonal characteristics are investigated in the next section 5, these sentences are deleted in the paragraph.

38. Line 331: ". We consider that winter covers the months of December, January and February, while March, April and May are spring, June, July and August are summer and the rest is autumn"

This sentence has been changed in the revised manuscript.

39. Line 333: "As we expected,.."

"As we expect" is corrected as "As we expected"

40. Line 334: "As the spot of direct sunlight slowly moves northward, the mean variance gradually increases" While looking at the figure, it is clear that not only the intensity of the variance increases but also and more importantly, the height up to which these large values are reached also increases, as well as the time duration of them. These facts should be included in the current analysis.

We agree with your suggestion. And we add the description as "moreover, the coverage height and time duration of its large values show an analogous monthly variation."

41. Line 335: "... August, and then decreases  gradually..."

"step by step" has been replaced with "gradually".

42. Line 336: "variance is significantly larger in spring than in autumn" Please be more specific, here you could add some numbers.

This sentence has been changed as

"the variance is significantly larger in spring than in autumn. Not only the maximum CBLH of 1.14 km at 13:30 in spring is much higher than that of about 0.66 km at 13:30

and 14:00 in autumn, but also the mean $\sigma_w^2$ of 0.42 m$^2$s$^{-2}$ in the CBL during spring is stronger than that of 0.35 m$^2$s$^{-2}$ during autumn."

43. Lines 336 and 337: " These monthly and seasonal features of convectively driven turbulence dominate the evolution of monthly and seasonal mean CBLHs." This is a sentence with high repetitiveness and not really with any information, please re-write it or don't include it.

    According to your suggestion, this sentence is removed in the revision.

44. Lines 342-345: Authors analyze figure 10, where maximum value of CBLH is presented for the 12 months. However it is not clear why is this maximum value chosen and there is a lack of analysis on the seasonality that this data implies. Furthermore, figure 10 also shows the local time when these maximum values were reached but this is not further analyzed nor explained, what does it imply? Then They start a sentence on line 344 saying "In weather forcast record there are 7,3,13,3 and 0 days with moderate to heavy rain..." But there is a lacking of a connecting sentence that connects this idea with the one in the previous sentence. Why are authors addressing heavy rain here? What do these numbers mean?

    As you said, the monthly and seasonal mean CBLHs are presented in Figure 9, and their maximum heights are chosen as Figure 10, which is repetitive. Hence, Figure 10 is deleted in the revised manuscript.

45. Lines 351-357: " As shown in Figure 9, the CBLH in July has the largest standard deviation (between 13:00 and 19:00) and the latest occurrence time of maximum value over the whole year, which is possibly attributable to the cloudy and rainy weather in addition to the strongest radiation. Similarly, the variability of weather conditions may be a major reason why the maximum height arises 1-2 hours earlier in April-June than in March. Nevertheless, with the gradual decline of solar radiation, the occurrence time of maximum height is steadily advanced from 17:30 in July to 13:00 in November and December." This explanation about figure 9 should be before in the Manuscript, before talking about figure 10, please move it and make it consistent.

Figures 10 is deleted, and the relevant description is shifted to the explanation about figure 9.

46. Lines 363-368: " The occurrence time of averaged maximum CBLH is the earliest at about 12:40 in December and the latest at 15:45 in August, which is slightly distinguished from those in the maximum value of mean CBLH. The standard deviation of occurrence time is obviously large in January, July and September. These results imply that the maximum height and its occurrence time of daily CBL are significantly influenced by the weather conditions besides radiation since the VV variance as a proxy of convectively driven turbulence is sensitive to the weather changes." These lines first are purely descriptive but I don't see any elucidating analysis on how this maximum CBLH at different times during different months are showing any new information. Again there is a lack of discussion about the processes that can be responsible for those variations. And when the authors mention the "weather conditions" they should be more specific i which conditions are they referring to and how do they change seasonally. Please re-write this making a valuable physical analysis or maybe even don't use the plots on figure 10 if they don't have any conclusive fact about it.

We agree with your comment. Indeed, Figures 10 and 11 come from Figure 9, and do not provide more new information, and are removed in the revision.

In particular, the retrieved CBLH is dominated by the surface heating associated with solar radiation, and is also significantly impacted by many factors, such as air humidity, ground moisture, clouds within the CBL and clouds topped the CBL, mid- and high-level clouds, and even sand and dust. Because of this, the CBLH may also display a large change from day to day even in a same month. Hence, it is relatively easy for a case in a specific day to clarify the influence of weather conditions on the CBLH. In the statistical analysis, the effects of weather conditions require lots of detailed weather data that we lack. To focus on the purpose, Figures 10 and 11 are deleted.

47. Lines 371-373: " In this study, we investigate the diurnal evolution of monthly and seasonal mean CBLH at Wuhan by the VV variance method based on the Ka-band MMCR observation,

and compare the CBLH evolution with that by the RCS gradient, variance and wavelet methods from the lidar measurement." You should add a sentence mentioning that also some study cases were investigated and why those particular study cases; how do they relate to your findings.

According to your suggestion, we add the description in Summary, as follows,

"Although the RCS is proportional to aerosol concentration and $w$ represents the vertical motion of aerosol particles, the comparison of four examples in different seasons indicates that the CBLHs from the MMCR $w$ are in good agreement with those from the lidar RCS, except for the initial growth and final decay phases. The discrepancy can mainly be attributed to the aerosol residual layer and the lidar blind range. The influence of residual layer on the lidar RCS generally causes an overestimation of CBLH, meanwhile, it is impossible for lidar to capture the CBL top within its large blind range. In addition, the CBLH in the MMCR observation shows less contamination by the long-range transport of sand and dust, and thick high-level clouds due to the rapid response of aerosol $w$ relative to its concentration. In this case, the MMCR observation can capture the diurnal evolution of CBLH."

48. Line 374: Be more specific when saying "statistically analyze" what do authors refer to by that?

"we statistically analyze the monthly and seasonal variations of CBLH." is rephrased as "we investigate the evolution of monthly and seasonal mean CBLHs".

49. Lines 379-381: "The occurrence times is between 13:00 in November and December and 17:30 in July for the maximum value of monthly mean CBLH, but between about 12:40 in December and 15:45 in August for the monthly mean value of daily maximum CBLH, respectively." Authors should explain why are they investigating these times and specifically metioning them even in the summary. I don't see any clear elucidating analysis about these times.

As Figures 10 and 11 are removed, the sentence is deleted in the revision.

50. Line 382: "  behavior, the seasonal mean CBLH has the maximum heights of 1.29 km at 14:30 and 15:00 in summer, 1.14"

"feature" has been replaced with "behavior".

51. Lines 383-384: "These results are similar to those in earlier studies..." Similar in what sense? The variables, seasons, comparison between methodologies and its particular characteristics or some more objective facts need to be specified in this sentence. Also maybe it is worth mentioning the influence of different terrain, latitudes, synoptic conditions. How is this comparable?

According to your suggestion, we make a careful comparison, which is moved to Section 5, as follows,

"In previous studies, based on the threshold of $\sigma_w^2$ from the Doppler lidar measurement in Mexico City (19.3° N, 99.1° E), the CBLH is higher in spring and summer, and lower in winter, while the maximum CBLH of about 1.5 km occurs in May, which is because the CBLH is suppressed to some extend by increased cloud cover in the rainy season between June and September (Burgos-Cuevas et al., 2021). However, the CBLH retrieved from the ceilometer backscatter data is obviously larger than that from the threshold of $\sigma_w^2$ (Burgos-Cuevas et al., 2021; Tang et al., 2016). Similarly, in the estimation of CBLH from the lidar RCS over Wuhan and Granada (37.18° N, 3.60° E), the maximum values of seasonal mean CBLHs in all the seasons are larger than those in our results although the gradual ascent of CBLH from winter and autumn to spring and summer is consistent with that in our results (Kong and Yi, 2015; Granados-Muñoz et al., 2012)."

52. Lines 391-392: " ...the CBLH from the lidar RCS is higher than from the MMCR VV variance, due to the high blind range of lidar and the strong influence of aerosol residual layer on the lidar RCS." It is not clear why is this due to the high blind range of lidar. Please explain further.

The description is rewritten as "it is impossible for lidar to capture the CBL top within

its large blind range."

53. Line 395: "Additionally, in comparison to the lidar RCS affected by the history of aerosol mixing processes, the CBLH" It is not clear how are the authors attributing this to the history of aerosol mixing processes. Please address it more specifically.

It should be "the change of aerosol concentration lags behind the variation of its vertical velocity". Because of the comparison is move to the second paragraph, "Additionally, in comparison to the lidar RCS affected by the history of aerosol mixing processes" is removed in the revised manuscript.

54. Line 407: Please specify what "weather conditions" in particular are the authors referring to. They should include maybe how was temperature, pressure, humidity, was there rain? How were the synoptic conditions on the investigated days. This implies that the authors may need to consider more information and make extra plots showing this or justifying it in a different manner.

We agree fully with your suggestion. The weather conditions mean to consider more information and make extra plots. The sentence is deleted, and the revised manuscript is more focused relative to our previous one. Finally, we really appreciate your many valuable comments and suggestions on our manuscript, again.

---

## Author Comment (AC2)

Response to Referee 2,

We sincerely thank you for your valuable comments and suggestions, which is of great help in improving our manuscript.

According to your suggestions, we revised our manuscript, and presented a point-to-point reply to your comments.

**Major comments:**

1. Too many expressions are used to describe the planetary boundary layer (PBL) in this manuscript, including boundary layer, boundary, CBLH. The PBL can be basically divided into convective boundary layer, neutral boundary layer and stable boundary layer, according the atmospheric static conditions. Therefore, one of my greatest concern is the topic of this work is the evolution of CBL height based on the measurements from MMCR. To my knowledge, the determination of CBL needs the temperature profiles. But I can not see any such profiles in the retrieval process of CBL height.

Sorry for our unclear expression. In the revised manuscript, only the PBL and CBL are presented, and the other expressions are changed.

We fully agree with you that the determination of CBL needs the temperature profiles. As you know, it is difficult to obtain the temperature profiles with high temporal resolution, thus we cannot compare the diurnal evolution of CBLH from the temperature measurement with that from the MMCR observation. Here, we compare the CBLH from the MMCR vertical velocity with that from the temperature in the routine radiosonde observation. The radiosonde is routinely released in Wuhan at 8:00 LT and 20:00 LT. The sun has gone down at 20:00, thus we present the comparison at 8:00.

The radiosonde data are provided by University of Wyoming at the website of https://weather.uwyo.edu/upperair/bufrraob.shtml. The Radiosonde data in Wuhan have a very rough vertical resolution of about 0.5-1.0 km before June 2021, and since then, the vertical resolution has been improved to about tens to hundreds of meters in the thousands of meters above the surface. Thus, we choose some days without precipitation in the second half of 2021.

We identify the CBLH by the maximum gradient of potential temperature ($\theta$) and a 0.25 threshold of bulk Richardson number ($Ri$) based on the radiosonde data (Guo et al., 2021; Seidel et al., 2010; Seibert et al., 2000). The bulk Richardson number is expressed (Guo et al., 2021; Seidel et al., 2010; Seibert et al., 2000) as follows:

$$Ri(z) = \frac{\left(\frac{g}{\theta_{vs}}\right)(\theta_{vz} - \theta_{vs})z}{(u_z - u_s)^2 + (v_z - v_s)^2 + (bu_*^2)},$$

where $g$ is the gravitational acceleration; $z$ is the height; $\theta_v$ is the virtual potential temperature; $u_*$ is the surface friction velocity; $u$ and $v$ is the zonal and meridional wind components, respectively; and $b$ is a constant, which is usually set to zero due to the fact that friction velocity is much weaker compared with the horizontal wind (Seidel et al., 2012). The subscripts of $z$ and $s$ denote the parameters at $z$ height and surface level, respectively.

Figure S1 (only presented in the Response) shows the comparison of CBLHs derived from the three methods at 8:00 in some days.

As shown in Figure S1, the CBLH from the threshold of $\sigma_w^2$ from MMCR is generally consistent with the CBLHs from the gradient of $\theta$ and the threshold of bulk $Ri$ from radiosonde, which indicates that the MMCR measurement can be used to estimate the CBLH in Wuhan.

Nevertheless, there is also a little difference in the results from the three methods, which is mainly because these methods have the different physical principles and mathematical algorithms. Similarly, a little difference can be seen in the CBLHs derived from an identical variable of lidar RCS by three methods.

[Figure]

**Figure S1.** Comparison of CBLHs estimated by the threshold of $\sigma_w^2$ from MMCR, and the maximum gradient of $\theta$ and threshold of Bulk $Ri$ from radiosonde data at 8:00. There are the three panels in each day, and the blue horizontal line denotes the position of estimated CBL top. In the panel (a), the gray thin and black thick lines denote the vertical velocity and their mean values from MMCR, respectively; the red line denote $\sigma_w^2$; and the yellow line denotes the threshold of $\sigma_w^2=0.3$ m$^2$ s$^{-2}$. In the panel (b), the black and red lines denote $\theta$ and its gradient from radiosonde, respectively. In the panel (c), the black line denotes $Ri$ from radiosonde data, and the yellow line denotes the threshold of $Ri=0.25$.

Sorry, this MMCR did not work routinely for many days or several months in 2021 due to the epidemic and the transmitter returned to the factory for maintenance, thus the MMCR data in only 2020 is used in the investigation.

In the revised manuscript, we add the sentence as follows,

"By comparison (not presented), the CBLH at 8:00 derived from the threshold of $\sigma_w^2$ =0.3 m$^2$ s$^{-2}$ is generally consistent with the CBLHs estimated by the maximum gradient of potential temperature and a 0.25 threshold of bulk Richardson number based on the radiosonde data at Wuhan, which indicates that the threshold of $\sigma_w^2$ =0.3 m$^2$ s$^{-2}$ is appropriate for the CBLH estimation at Wuhan."

2. My second concern lies with the physical basis for the PBL height retrieval from MMCR. As the authors stated in the Introduction section, "there are few reports on the use of vertical velocity obtained from Doppler cloud radar for the CBL investigations." To the best of my knowledge, the cloud-topped PBL is extremely complex due to the complicated turbulence-convection interaction, and the entrainment/detrainment process near the cloud edges. Nevertheless, the MMCR can not efficiently obtain any information (e.g., the vertical velocity) in the absence of cloud, which exactly corresponds to the cloud-topped PBL. Even though the authors say that there exists a weak echo layer near the surface, this could be due to the clutters. If not, the PBL top is well above the near surface layer. Then I pose a question "what is the physical basis for the CBLH and how reliable?"

We think that this is a very good question, and we also thought about the question of physical basis carefully before we did the work. We would like to explain it from the following aspects.

Firstly, conventional microwave radar is pulse radar, and the MMCR in our work is a frequency-modulated continuous wave (FMCW) radar. Due to the continuous transmitting and receiving, the FMCW radar has much larger mean power relative to pulse radar, which is why the MMCR in the manuscript can always receive weak echoes from aerosols near the ground.

Secondly, we analyze the FMCW radar sensitivity, which refers to the minimum reflectivity detected by radar at a certain altitude. We calculate the radar sensitivity according to the radar equation, which is shown in Figure S2, together with the distribution of reflectivity from the MMCR observation in June 2020. The number of

reflectivity count is within the bin of 30-m height (a radar range gate) and 1-dBZ reflectivity. The minimum detectable reflectivities at different heights are close to the calculated value, indicating the fine performance of MMCR.

The high occurrence rate of reflectivity between -40 and -60 dBZ below 2.5 km is due to various large aerosol particles, including low-level cloud particles (Browning and Atlas, 1966; Chandra et al., 2010; Moran et al., 1998; Zhang et al., 2024).

We calculate the total number of reflectivity count at each radar range gate (30 m height) in June 2020, which is shown in Figure S3. Below about 1.8 km, the total number at each height is more than $4.3 \times 10^4$, approximately equal to $4.32 \times 10^4$ (June has $60 \times 24 \times 31 = 4.32 \times 10^4$ min). The MMCR has a 5-min self-check every day, which indicates no recorded data for 155 min in June. Hence, this means that there always exists echo signal below 1.8 km in June. The low-level clouds from the surface to 1.8 km cannot occur throughout June, thus the weak echoes from the surface to about 1.8 km come mainly from the backscattering of various large aerosol particles, including low-level cloud particles.

[Figure]

**Figure S2**. (red line) Sensitivity calculated vs. (color shading) statistical number of reflectivity from MMCR observation in June 2020.

[Figure]

**Figure S3**. Total number of reflectivity count from MMCR observation in June 2020.

Thirdly, in previous studies, the weak echoes near the surface are attributed to the backscattering of small insects and aerial plankton in some studies (Franck et al., 2021; Chandra et al., 2010; Achtemeier, 1991), and are also suggested to come from the scattering of dust particles in other studies (Görsdorf et al., 2015; Clothiaux et al, 2000; Moran et al., 1998).

Aerosol particles typically range in size from 0.01 to 10 μm, with floating dust particles occasionally reaching 10 μm in diameter. Smoke particles are typically less than 1 μm in size, while plant aerosols range in size from 5 to 100 μm, and combustion-generated aerosols exhibit a particle size of 0.01 to 1000 μm. For cloud particles with diameter $D$=10 μm, the number concentration of 100 $cm^{-3}$ have a reflectivity factor of $Z$= -40 dBZ. Assuming a same dielectric constant, a reflectivity factor of $Z$= -50 dBZ corresponds to the number concentration of 10 $cm^{-3}$ for $D$=10 μm aerosol particles. Therefore, considering that there are lots of larger aerosol particles in the lower atmosphere, for example, dust particles visible to the eyes in the sunlight coming in through the window, various large aerosol particles (including not only low-level cloud particles, but also plant aerosol particles, dust particles, and all kinds of urban emission particles) can cause a very weak reflectivity (-60 to -40 dBZ) in the MMCR observation, as shown in the

statistical result from Figure S2, in other words, the MMCR can obtain the reflectivity and vertical velocity information through aerosol particle backscattering in the absence of low-level clouds.

Finally, as you said, the cloud-topped PBL is extremely complex due to the complicated turbulence-convection interaction, and the entrainment/detrainment process near the cloud edges. We present three examples of clouds in Figures S4-S6. There are mainly the low-level (cloud-topped), mid-level and high clouds in Figures S4, S5 and S6. In all three cases, the CBLH can be identified by the vertical velocity variance, moreover, the clouds in different heights have the influence on the CBLH. There are the complicated dynamic processes involved in the interaction between the PBL and the clouds, thus the MMCR observation provides an effective means to explore these complex processes since the MMCR is a powerful instrument for observing clouds and weak precipitation, and this is also what we will continue to strive for in the future.

[Figure]

**Figure S4**. (a) Reflectivity, (b) vertical velocity and (c) vertical velocity variance from MMCR observation on 20 February 2020. The black line denotes the identified CBLH.

[Figure]

**Figure S5**. Same as Figure S4 but for 14 April 2020.

[Figure]

**Figure S6**. Same as Figure S4 but for 7 June 2020.

**Specific comments:**

1. Lines50-51: it is totally wrong to state that "In the afternoon, ***turbulent activity is weakened". Conversely, in the absence of cloud or synoptic-scale weather system, the turbulence tends to reach the maximum, due to the strongest sensible heat flux in the afternoon.

We agree fully with you. It should be "In the late afternoon". Combing your suggestion with that from the other Referee, this sentence is corrected in the revision as,

"On a clear day, the CBLH rises after sunrise and reaches its maximum in the afternoon."

2. Lines 51-52: Except for the existence of aerosol particles in the residual layer, most of them are present in the nocturnal stable PBL.

According to your suggestion, we rephrase the sentence to be,

"most of aerosol particles within the CBL are deposited into the nocturnal stable PBL due to the rapid weakening of convectively driven turbulence, and some particles are transformed into an aerosol residual layer."

3. Line 68: What is the "the boundary top"? is it different from the atmospheric boundary layer top? If not, why not use the same term throughout the whole manuscript?

"the boundary top" is changed as "the CBL top".

According to your helpful suggestion, the boundary layer (top or height) in the manuscript is concretized as the PBL or CBL (top or height).

4. Lines 76-77: "The radiosonde data have a widely geographical distribution and long-term accumulation" should be rephrased.

Combing the other Referee' suggestion, we delete this sentence in the revised manuscript.

5. Line 78: "boundary layer" refers to "planetary boundary layer height ()"?? if so, what are the difference between PBLH and CBLH?

In this paragraph, "boundary layer (height)" is reworded as "CBL (CBLH)".

6. Line 92-96: Too long sentence for "Wind profile radar can … turbulence (Liu et al., 2020…"
and thus I suggest to rephrase it.

In the revision, the sentence is divided into two sentences, as follows,

"The vertical gradients in temperature, humidity and turbulence change the profile of
atmospheric refractive index, which can cause the scattering of electromagnetic waves.
Wind profile radar obtains the atmospheric wind speed and direction by decomposing
the Doppler shift of backscattered waves."

7. Line 109: I do not understand the meaning of "Plunge".

In the revised manuscript, "by tracing the height of aerosol concentration plunge" is
changed to be,

"by tracing the height where the aerosol concentration sharply decreases with height."

8. Line 114: "capability" -> "incapability"

"limited by the ability of lasers to penetrate clouds" is corrected as "due to the
incapability of lasers to penetrate clouds".

9. Line 121: Is there a rapid decline stage of CBL in the afternoon?? If so, some necessary
references are needed to be provided here.

Sorry for our unclear description. "the rapid decline stage of CBL in the afternoon" is
addressed to be,

"the rapid decline stage of CBL in the late afternoon (Dewani et al., 2023; Manninen et
al., 2018; Schween et al., 2014; Barlow et al., 2011)".

10. Line 123: Please elaborate on the definition of "historic effect".

In the revised manuscript, "This discrepancy often reflects the historical effect of aerosol
mixing rather than the current situation of convectively driven turbulence" is rephrased
to be,

"This discrepancy is due to aerosols from a long time-mixing process rather than the
current situation of convectively driven turbulence".

11. Language: There are too many grammar errors or inappropriate expression throughout the whole manuscript. I can not continue the reviewing processes if the authors did not seek help from a native English editor or colleague.

According to your constructive suggestion, we have conducted a thorough review of the paper and sought assistance from English-speaking colleagues to correct grammatical errors and improve expressions. And we thank you very much for your valuable comments and suggestions on our manuscript, again.

---

## Referee Report (RR1)

**Technical note: Evolution of convective boundary layer height estimated by Ka-band continuous millimeter wave radar at Wuhan in central China**

Zirui Zhang[1,2,3]  Kaiming Huang[1,2,3]  Fan Yi[1,2,3]  Fuchao Liu[1,2,3]  Jian Zhang[4]  and

Yue Jia[5]

[1]School of Electronic Information, Wuhan University, Wuhan, China

[2]Key Laboratory of Geospace Environment and Geodesy, Ministry of Education, Wuhan, China

[3]State Observatory for Atmospheric Remote Sensing, Wuhan, China

[4]School of Geophysics and Geomatics, China University of Geosciences, Wuhan, China

[5]NOAA Chemical Sciences Laboratory, Boulder, CO, USA

Email Address: hkm@whu.edu.cn

Zip code: 430072

**Abstract.** Using the vertical velocity ($w$) observed by a Ka-band millimeter wave cloud radar (MMCR)

at Wuhan, we investigate the evolution of convective boundary layer height (CBLH) based on a specified

*Better say: "When compared with a methodology based on RCS..."*

threshold of vertical velocity variance ($\sigma_w^2$). By comparison, the MMCR-derived CBLH is generally in good agreement with that retrieved from the lidar range corrected signal (RCS), except for a few hours post-sunrise and pre-sunset due to the influence of aerosol residual layer on the lidar RCS. Meanwhile, the

*what does contamination means in this context?*

CBLH estimated from the MMCR $\sigma_w^2$ shows less contamination by passing sand and dust, and thick

*better say: "...shows less variations attributted to the noise due to sand and dust"*

high-level clouds due to the rapid response of aerosol $w$ relative to its concentration, thus the MMCR

measurement can capture the diurnal evolution of CBLH. The MMCR observation in 2020 depicts the

*Better say: "The diurnal and seasonal evolution of the CBLH is investigated via MMCR measurements"*

diurnal evolution of seasonal and monthly mean CBLHs. The seasonal mean CBLH reaches the peak heights of 1.29 km in summer, 1.14 km in spring, 0.66 km in autumn, and 0.6 km in winter, indicating the

*better say: "which is due to the stronger surface heating due to higher radiation in summer" And this is not a new discovery, it is well-known*

dominant effect of radiation heating. The maximum value of monthly mean CBLH rises steadily from 0.66

km in January to 1.47 km July, followed by a gradual decline to 0.42 km in December. Statistical standard

*This is also well-known. What is the novelty of the paper? that is what should be on the abtsract*

[revised manuscript text omitted]

---

## Author Response (AR2)

Dear editor,

We sincerely appreciate you and the anonymous reviewers for valuable comments and suggestions on our manuscript, again.

We have carefully considered these comments and suggestions, and have revised our manuscript accordingly.

The point-by-point responses to the reviewers are provided, along with the revised manuscript showing tracked changes.

With best regards,

Huang Kaiming

Response to Referee #1,

We sincerely appreciate your insightful comments and suggestions, which have greatly contributed to improving our manuscript.

According to your suggestions, we revised our manuscript, and presented a point-to-point reply to your comments.

**Comments.**

1. On my former first general comment where I pointed out that the focus and aim of the paper should be more clear, authors mention that it is addressed in the last paragraph of section 1; and it is more explained there, but still a further explanation is necessary.

Combining the suggestion in PDF, we rewrite the Abstract in the revised manuscript, as follows,

"Using the vertical velocity ($w$) observed by a Ka-band millimeter wave cloud radar (MMCR) at Wuhan, we investigate the evolution of convective boundary layer height (CBLH) based on a specified threshold of vertical velocity variance ($\sigma_w^2$). The CBLHs from the MMCR $w$ in the selected durations are compared with those estimated by the lidar range corrected signal (RCS) and radiosonde temperature based on different algorithms, showing good agreement with each other. Although these algorithms are on basis of different dynamic and thermodynamic effects, the diurnal evolution of CBLH from MMCR is generally consistent with that from lidar, except for a few hours post-sunrise and pre-sunset due to the influence of aerosol residual layer on the lidar RCS. Meanwhile, the CBLH from MMCR shows less variation in occurrence of sand and dust, and swifter response of thick clouds, relative to that from lidar. In this case, $\sigma_w^2$ of the MMCR $w$ identifies the CBLH based on dynamic effect, which can accurately capture the diurnal evolution of CBLH compared with that from the change of long time-mixing aerosol concentration. The monthly and seasonal features of CBLH at Wuhan is revealed via the MMCR measurement. Hence, considering that the MMCR is capable of continuous observation in various weather conditions, the MMCR $w$ with high resolution can be applied to monitoring the evolution of CBLH in different atmospheric conditions, which is helpful for improving our comprehensive understanding of CBL and dynamic processes in CBL."

And in the last paragraph of Introduction, we add the explanation as,

"In present study, we estimate the CBLH based on the vertical velocity from a Ka-band millimeter wave cloud radar (MMCR) at Wuhan, and compared this result with that derived from the lidar RCS by three

algorithms, and from radiosonde data by two algorithms. These algorithms are on basis of different dynamic and thermodynamic effects, respectively, thus the comparison enhances our comprehensive understanding of CBL and retrieval algorithms."

2. When they say "algorithms" in line 122. There are algorithms based on thermodynamic characteristics (i.e. profiles of temperature and/or humidity), on dynamic characteristics (wind, turbulence) and on backscatter. A review of these different methodologies can be found in Kotthaus et al. (2023) (https://doi.org/10.5194/amt-16-433-2023). The algorithm utilized by the authors is clearly a dynamical one but that should be clearly stated and the capabilities and limitations that this approach has needs to be considered. Moreover it is compared with the RCS which as far as I understand is a backscatter-based algorithm. All this should be clearly assessed while the authors compare their methodologies.

According to your suggestion, we have added the description of algorithm in the last paragraph of Introduction as,

"In present study, we estimate the CBLH based on the vertical velocity from a Ka-band millimeter wave cloud radar (MMCR) at Wuhan, and compared this result with that derived from the lidar RCS by three algorithms, and from radiosonde data by two algorithms. These algorithms are on basis of different dynamic and thermodynamic effects, respectively, thus the comparison enhances our comprehensive understanding of CBL and retrieval algorithms."

And in Section 3.2 in the revised manuscript, we have added the capabilities and limitations of methods, as follows:

"$\sigma_w^2$ indicates the turbulence level under the current condition, whereas, RCS tends to reflect the variation in the concentration of long time-mixing aerosol particles caused by dynamic effects (Kotthaus et al., 2023). Hence, the threshold method is a dynamical algorithm, which is more effective in capturing the dynamic changes within the CBL compared to the aerosol concentration algorithm based on the lidar RCS. In this way, the MMCR observes the high-temporal resolution data of $w$, making it available for analyzing diurnal evolution of CBL in different months and seasons. However, based on earlier studies, the selected threshold values are subject to change across the different regions (Burgos-Cuevas et al., 2023; Schween et al., 2014; Pearson et al., 2010; Tucker et al., 2009)."

The added literature of Kotthaus et al. (2023) is listed in References.

3. The introduction is less repetitive and better structured as was asked by me, however still some asseverations should be more precisely stated, such as the ones pointed out in lines 66 and 80 (please see the corrected attached pdf for this).

Sorry, we couldn't find the specific comments at lines 66 and 80 in the PDF.

According to your suggestion, we added a detailed explanation for the two sentences, as follows,

"Radiosonde can obtain high-precision meteorological parameters, such as temperature, humidity, horizontal wind and pressure, providing the possibility of estimating CBLH through various algorithms",

And "Wind profile radar can measure the atmospheric wind speed and direction by analyzing the Doppler shift of the backscattered waves of multiple beams. The electromagnetic beams are reflected back due mainly to the atmospheric refractive index change caused by the non-uniform vertical structure of the atmosphere, such as vertical gradients in temperature, humidity, and turbulence, thus received echo and retrieved wind from radar contain the information related to the atmospheric vertical structure."

4. I had some comments in the abstract before, and changes have been made, however, it is still not accurate enough. Please see the specific comments on the abstract added to the pdf revised version and make the corresponding changes.

According to your suggestion, the Abstract is rewritten in the revision, and please see the Comment 1.

5. However, as a general comment on the abstract I want to highlight the fact that it barely deals with novel aspects of the paper. It does state that there are higher CBL heights coincident when there is stronger radiation, but that is a very well-known fact, so the authors should re-structure the abstract and stress more on what is actually a scientific novelty: the evaluation of the vertical velocity from the MMCR to investigate the CBL height. They should look into capabilities and limitations of this approach. And if it is capable of assessing diurnal and seasonal variabilities then say that, but do not state the sentences as if the novelty was that higher CBLHs correspond to higher radiation, because that is not a novelty.

In the revised abstract, we have restructured it to highlight the use of vertical velocity derived from MMCR to estimate CBLH:

"The results are compared with those estimated by range corrected signal (RCS) from lidar and

temperature from radiosonde based on different algorithms. Although these algorithms are on basis of different dynamic and thermodynamic effects, the diurnal evolution of CBLH from MMCR is general agreement with that from lidar, except for a few hours post-sunrise and pre-sunset due to the influence of aerosol residual layer on the lidar RCS. Meanwhile, the CBLH from MMCR $\sigma_w^2$ shows less variation in occurrence of sand and dust, and swifter response of thick high-level clouds, relative to that from lidar. In this case, $\sigma_w^2$ of MMCR $w$ identifies the CBLH based on dynamic effect, which can accurately capture the diurnal evolution of CBLH compared with that estimated from the change of long time-mixing aerosol concentration in lidar observation. The monthly and seasonal features of CBLH at Wuhan is revealed via MMCR measurement. Hence, considering that the MMCR is capable of continuous observation in various weather conditions, the MMCR $w$ with high resolution can be applied to monitoring the evolution of CBLH in different atmospheric conditions, which improves our comprehensive understanding of CBL and the dynamic processes in CBL."

6. In the methodology I had pointed out that the specifications of the radar were missing and still this is the case for some, as it is pointed out in the pdf, please check specific comments there.

The MMCR is designed by the Atmospheric Remote Sensing Observatory (ARSO) of Wuhan University (WHU). Transmitter, antenna, servo, data processor and cabin are purchases from different manufacturers, respectively. The integration of radar system is carried out by the combination of Nanjing Industrial Company and WHU ARSO. The radar uses a continuous wave (CW) system, call WHU-CW MMCR. This radar was established by the ARSO in 2019. Now, we can synthesize a larger power of 60-80 W for transmitter, improving further the detection capability of radar.

In the revised manuscript, the sentence is rewritten as,

"The WHU-CW MMCR established by the ARSO adopted a continuous wave (CW) system, and is a Ka-band frequency-modulated continuous wave (FMCW) Doppler radar. The MMCR is installed in WHU, as shown in Figure 1."

7. In order to evaluate how well their vertical variance threshold methodology acts to estimate CBLH the authors compare with radiosonde at 0800 LT. This is valuable, however discussion is missing. Please answer these questions:

(1). What does "generally consistent" (line 238) mean in terms of standard deviation? or some other objective

quantification that you should utilize to quantitatively, statistically and significantly compare both methodologies.

In the revised manuscript, we add the comparison of CBLHs from MMCR and radiosonde, and present the added figures as Figures 5 and 6.

The added description and figures are following as,

"$\sigma_w^2$ of MMCR $w$ determines the CBL top from the perspective of dynamic effect, and the CBLH can be estimated from the temperature data based on the thermodynamic effect. Here, we compare the CBLH derived from the MMCR $w$ with that from the radiosonde data. Radiosonde is typically launched in Wuhan at 08:00 and 20:00. Given that the sun has set by 20:00, we present the comparison at 08:00. The radiosonde data are provided by University of Wyoming from the website at https://weather.uwyo.edu/upperair/bufrraob.shtml. The vertical resolution of radiosonde data in Wuhan was approximately 0.5-1.0 km before June 2021, and then was improved to a range of tens to hundreds of meters at higher altitudes. Therefore, we select the high-resolution data in the days without precipitation for our analysis.

We estimate the CBLH from the radiosonde data by using the methods of potential temperature ($\theta$) gradient and bulk Richardson number ($Ri$) threshold. The potential temperature gradient ($Grd_\theta$) is calculated at two adjacent heights in the radiosonde data, and the CBLH is determined by the maximum gradient in the profile of $Grd_\theta$ (Seidel et al., 2010). The bulk Richardson number is expressed (Zhang et al., 2014; Seibert et al., 2000), as follows,

$$Ri(z) = \frac{(g/\theta_{vs})(\theta_{vz} - \theta_{vs})z}{(u_z - u_s)^2 + (v_z - v_s)^2 + (bu_*^2)} \tag{1}$$

where $g$ is the acceleration due to gravity; $z$ is the height; $\theta_v$ is the virtual potential temperature; $u_*$ is the surface friction velocity; $u$ and $v$ are the zonal and meridional wind components, respectively; and $b$ is a constant, which is usually set to zero due to the fact that friction velocity is much weaker compared with the horizontal wind (Seidel et al., 2012). The subscripts of $z$ and $s$ denote the parameters at $z$ height and surface level, respectively. In the profile of $Ri$, the CBLH is identified when $Ri$ firstly crosses a threshold value upward from the ground, and the threshold is typically taken as 0.25 in early studies (Guo et al., 2021; Seibert et al., 2000), which is chosen in the analysis.

Figure 5 shows the comparisons of CBLHs derived from the MMCR and radiosonde measurements at 8:00 on 21 and 25, July 2021, respectively. On 21, for a threshold of $\sigma_w^2$=0.3 m$^2$ s$^{-2}$, the CBLH of 0.39 km from the MMCR $w$ is in agreement with that of 0.40 km from the radiosonde $Grd_\theta$, which are slightly larger than that of 0.34 km from the radiosonde $Ri$. In contrast to this, on 25, the CBLH is 0.57 km from the MMCR $w$, which is consistent with that of 0.59 km from the radiosonde $Ri$, but is slightly higher than that of 0.45 km from the radiosonde $Grd_\theta$. Nevertheless, in the whole, the results from all the three methods roughly agree with each other.

[Figure]

**Figure 5.** Comparison of CBLHs estimated by (a, d) threshold of $\sigma_w^2$=0.3 m$^2$ s$^{-2}$ from MMCR, and (b, e) maximum gradient of $\theta$ and (c, f) threshold of $Ri$=0.25 from radiosonde data at 08:00 on (upper) 21 and (lower) 25 July 2021. In the panels 5a and 5d, the gray and black lines denote (lower horizontal axis) $w$ and its mean value from MMCR, respectively, and the red and yellow lines denote (upper horizontal axis) $\sigma_w^2$ and the threshold of $\sigma_w^2$=0.3 m$^2$ s$^{-2}$, respectively. In the panels 5b and 5e, the

black and red lines denote (lower horizontal axis) $\theta$ and (upper horizontal axis) its gradient from radiosonde, respectively. In the panels 5c and 5f, the black and yellow lines denote $Ri$ and the threshold of $Ri$ =0.25 from radiosonde data, respectively. The blue horizontal line represents the position of identified CBL top.

Figure 6 displays the scatterplot of CBLHs identified by the MMCR $w$, and the radiosonde $Grd_\theta$ and $Ri$ at 8:00 on the clear days in June and July 2021. The different variables and algorithms are used in the three methods, thus there are some differences of CBLHs derived from these methods, as shown in Figure 6. The CBLH from $\sigma_w^2$ of MMCR $w$ has the correlation coefficients of 0.83 and 0.81 with that from the radiosonde $Grd_\theta$ and $Ri$, respectively, which are highly consistent with the correlation coefficient of 0.83 from the radiosonde $Grd_\theta$ and $Ri$. These results support the threshold of $\sigma_w^2 = 0.3$ m$^2$ s$^{-2}$ applied to the CBLH estimation in Wuhan. In following analysis, we take 0.3 m$^2$ s$^{-2}$ as the threshold to determine the CBLH in the MMCR observation.

It can be noted that the comparison focuses solely on the CBLH at 8:00 rather than the diurnal evolution of CBLH, owing to the lack of radiosonde observation. Consequently, we analyze the diurnal evolution of CBLH derived from the MMCR and lidar measurements."

[Figure]

**Figure 6.** Scatterplot of CBLHs derived from (a) threshold of $\sigma_w^2$ =0.3 m$^2$ s$^{-2}$ from MMCR vs. maximum gradient of $\theta$ from radiosonde, (b) threshold of $\sigma_w^2$=0.3 m$^2$ s$^{-2}$ from MMCR vs. threshold of $Ri$ =0.25 from radiosonde, and (c) threshold of $Ri$ =0.25 vs. maximum gradient of $\theta$ from radiosonde.

(2). How does the evolution of the diurnal cycle affects this comparison? At 8:00 am the convection in the

ABL should be starting to take place, so this corresponds to a growing boundary layer phase, therefore it is a good time to estimate it with the variance as you do it, that makes sense and the utilization of radiosondes is also valuable. But you need to discuss that this is a phase in which CBL is growing and it will grow more during the diurnal cycle, however I assume that there are no later radiosondes to compare with a more developed boundary layer, which is ok but also needs to be stated.

According to your suggestion, we have added the clarification in the revision as,

"It can be noted that the comparison focuses solely on the CBLH at 8:00 rather than the diurnal evolution of CBLH, owing to the lack of radiosonde observation. Consequently, we analyze the diurnal evolution of CBLH derived from the MMCR and lidar measurements."

(3). Authors utilize now also the threshold of the bulk Richardson number to estimate a boundary layer height. This is a valuable methodology but needs to be much further described. The equation of the Richardson number is added in the answers but not in the paper, please add it to the paper and explain how the calculations were made. Furthermore, the difference between gradient and bulk Richardson number should be clear, the equation written in the answers is correct but the explanation lack the fact that the potential temperature at each height and at the surface is considered (difference with the surface and not a continuous derivative as in the gradient).

According to your suggestion, we have added the explanation of calculation method, and please see the response in Comment 7(1).

8. I had previously a comment on what do authors mean by "subsidence" and now they changed it to "dip" answering to my comment. However, they still did not explain what do they refer when they now use "dip" please explain. In my perspective this is even more unclear now. Therefore, it is highly recommended to be changed in the paper. I think that the authors mean that the CBL height reduces, but I am really not sure and it is not clear in the paper what they refer to nor if there would be a physical mechanism explaining this.

Sorry for the unclear expression. You are right. We have replaced three "dip" with "obvious reduction", "evident reduction" and "reduction" in the revision, respectively.

9. On line 151 you say: "...a maximum unambiguous velocity of..." and it is not clear if you are referring to an uncertainty of the velocity or what is it, please be more precise and use scientific vocabulary.

According to your suggestion, "a maximum unambiguous velocity of 4.30 m s$^{-1}$" was replaced with "a

maximum measurable velocity of 4.30 m s$^{-1}$ without aliasing effect".

Finally, we have made the correction about your suggestions presented in the supplementary PDF, and thank you for your careful and valuable suggestions on our manuscript, again.

Response to Referee #2,

  We are grateful to your suggestion and recognition for our work. We have made correction in response to your valuable suggestion.

Comment.

1.The authors have adequately addressed all my concerns in my previous review. in this revision, the quality of this revision is much improved and I appreciate their efforts very much. Therefore, I am pleased to recommend its acceptance for publication at ACP. There is one in-text citation that is incorrect in line 81: Liu et al., 2020 can be revised to "Liu et al., 2019 (doi:10.1109/TGRS.2019.2918301)".

  According to your suggestion, "Liu et al., 2020" has been corrected as "Liu et al., 2019" in the revision.

  Thanks!